# Cationic crosslinked carbon dots-adjuvanted intranasal vaccine induces protective immunity against Omicron-included SARS-CoV-2 variants

Hong Lei[1,4], Aqu Alu[1,4], Jingyun Yang[1,4], Xi He[1,4], Cai He[1], Wenyan Ren[1], Zimin Chen[1], Weiqi Hong [1], Li Chen[1], Xuemei He[1], Li Yang[1,2], Jiong Li[1,2], Zhenling Wang[1,2], Wei Wang [1,2], Yuquan Wei[1,2], Shuaiyao Lu [3] ✉, Guangwen Lu [1,2] ✉, Xiangrong Song [1,2] ✉ & Xiawei Wei [1,2] ✉

Mucosal immunity plays a significant role in the first-line defense against viruses transmitted and infected through the respiratory system, such as SARS-CoV-2. However, the lack of effective and safe adjuvants currently limits the development of COVID-19 mucosal vaccines. In the current study, we prepare an intranasal vaccine containing cationic crosslinked carbon dots (CCD) and a SARS-CoV-2 antigen, RBD-HR with spontaneous antigen particlization. Intranasal immunization with CCD/RBD-HR induces high levels of antibodies with broad-spectrum neutralization against authentic viruses/pseudoviruses of Omicron-included variants and protects immunized female BALB/c mice from Omicron infection. Despite strong systemic cellular immune response stimulation, the intranasal CCD/RBD-HR vaccine also induces potent mucosal immunity as determined by the generation of tissue-resident T cells in the lungs and airway. Moreover, CCD/RBD-HR not only activates professional antigen-presenting cells (APCs), dendritic cells, but also effectively targets nasal epithelial cells, promotes antigen binding via sialic acid, and surprisingly provokes the antigen-presenting of nasal epithelial cells. We demonstrate that CCD is a promising intranasal vaccine adjuvant for provoking strong mucosal immunity and might be a candidate adjuvant for intranasal vaccine development for many types of infectious diseases, including COVID-19.

Since December 2019, severe acute respiratory syndrome coronavirus 2 (SARS-CoV-2) has spread rapidly around the world and caused the coronavirus disease 2019 (COVID-19) pandemic[1]. The recently emerged Omicron variant has become the globally dominant strain[2], which has severely compromised the efficacy of currently available vaccines as mutations in the receptor-binding domain (RBD) confer these variants of concern (VOCs) the ability to escape immune attacks[2–4]. Therefore, next-generation vaccines that could provide

[1]Laboratory of Aging Research and Cancer Drug Target, State Key Laboratory of Biotherapy and Cancer Center, National Clinical Research Center for Geriatrics, West China Hospital, Sichuan University, 610041 Chengdu, China. [2]WestVac Biopharma Co. Ltd., Chengdu, China. [3]National Kunming High-level Biosafety Primate Research Center, Institute of Medical Biology, Chinese Academy of Medical Sciences and Peking Union Medical College, Yunnan, China. [4]These authors contributed equally: Hong Lei, Aqu Alu, Jingyun Yang, Xi He. ✉e-mail: lushuaiyao-km@163.com; lugw@scu.edu.cn; songxr@scu.edu.cn; xiaweiwei@scu.edu.cn

strong and broad protection are needed[5–7]. Recently, our group successfully produced and evaluated RBD-HR as a SARS-CoV-2 antigen, which takes advantage of the automatic assembly properties of the heptad-repeat sequences 1 (HR1) and HR2 on the spike protein of SARS-CoV-2[5]. The HR-fused recombinant RBD protein containing L452R and T478K mutations successfully self-assembled into trimers that showed excellent immunogenicity and induced a potent immune response against the Omicron variant with MF59 adjuvant after intramuscular immunization. However, the traditional immunization route is suboptimal in inducing mucosal immune responses, which are essential for defending against mucosally transmitted diseases, including SARS-CoV-2[8,9].

To date, great efforts have been made to develop intranasal COVID-19 vaccines, four of which have reached phase 3 clinical trials[10]. Recombinant protein-based vaccines have been extensively studied in the context of intranasal delivery[11,12]. However, due to the poor immunogenicity, the addition of adjuvants to recombinant protein-based vaccines is crucial to improve the persistence, breadth, and strength of immune responses[13]. Various adjuvants have been engineered for intranasal vaccines, including toxoid adjuvants (CTA1DD and LThaK), agonists of pattern recognition receptors (e.g., Poly I:C, monophosphoryl lipid A (MPL), and c-di-AMP), cytokines, etc.[8]. Despite promising efficacy, these adjuvants have not been approved for human use largely because of safety concerns.

Carbon dots (CDs), first discovered in 2004, have developed rapidly in the biomedical field due to their small size, controllable surface chemistry, excellent water dispersibility, good biocompatibility, and less environmental impact[14,15]. Because of incomplete carbonization, some functional groups can remain in the structure of CDs. This endowed CDs with unique properties from precursors and allowed them to be widely used to deliver DNA/RNA for gene therapy[16] and chemotherapeutic drugs[17]. The nano-scaled size has also inspired the application of CDs in delivering the model antigen ovalbumin (OVA) intranasally[14]. Nevertheless, the application of CDs in intranasal vaccines in a real disease state has not been verified yet.

In this study, we synthesized cationic crosslinked carbon dots (CCDs) using a linker to connect CDs through a simple ring-opening reaction, which can attract RBD-HR antigens with anionic polypeptides and form nanoparticles by electrostatic interactions. Intranasal vaccination with CCD-adjuvanted RBD-HR recombinant protein yielded robust, broadly protective, and long-term immune responses against SARS-CoV-2 infection, including Omicron variants. We discovered that in addition to dendritic cells (DCs), a classic professional antigen-presenting cell (APC), nasal epithelial cells (NECs) are also important for the presentation of CCD/RBD-HR for T-cell activation. Importantly, we revealed the underlying mechanism by identifying sialic acid (Neu5Ac) as a key molecule that mediates the binding of intranasally delivered antigens to NECs in the presence of CCD.

## Results

### CCD/RBD-HR nanoparticles

CCD was mixed with the RBD-HR antigen proteins for intranasal delivery (Fig. 1a). In this work, PEI 1.8 kDa was used as the precursor to fabricate CD-1.8k through the solvothermal method. Then, CCD was prepared with a linker to polymerize CD-1.8k via the ring-opening reaction. The hydroxyl group introduced by the linker could improve the antiserum ability of CCD to prolong its blood circulation time[18]. In the 1H NMR spectrum of CCD, the peaks ranging from 3.75–4.00 ppm indicated the existence of the introduced hydroxyl group (Supplementary Fig. 1). The broadband absorption at ~3400 cm$^{-1}$ of the CCD FT-IR spectrum suggested the existence of −OH and N−H groups (Fig. 1b). X-ray photoelectron spectroscopy (XPS) showed that C=O and N−H groups were present in the structure of CCD, consistent with the FT-IR results (Fig. 1c). The above results showed that the abundant hydroxyl groups present in the chemical structure endowed CCD with

good hydrophilicity and improved its antiserum ability to prolong the blood circulation time[18], while the amine group provided a positive surface charge for CCD. Transmission electron microscopy (TEM) showed that due to the cross-linking of the linker compound, the CCD particles tended to aggregate, resulting in a larger particle size than CD (Fig. 1d). Lysosomal escape is a rate-determining step in the delivery of nanoparticles. Thus, the acid−base titration experiment was carried out to investigate the buffer capacity of CCD in lysosomes (pH ~ 4) and the physiological environment (pH ~ 7.4) (Fig. 1e). In the pH range of 4.0 ~ 7.4, the NaOH consumption of CCD is higher than that of PEI 1.8k, indicating a superior buffering capacity of CCD, which can produce a strong proton sponge to promote lysosomal escape[19].

Our group successfully expressed the SARS-CoV-2 antigen recombinant trimer protein RBD-HR, the surface charge of which was determined to be approximately −8 mV (Fig. 1f). We found that due to the presence of the residual amino group, the zeta potential of CCD was highly positive: + 32 mV, compared to +20 mV of the CCD/RBD-HR mixture (Fig. 1f). With the mass ratio of CCD: RBD-HR increasing, CCD could encapsulate RBD-HR into stable and smaller nanoparticles (Fig. 1g). The particle size of CCD/RBD-HR (10:1) is ~60 nm, compared to <20 nm of CCD/RBD-HR (30:1), the latter of which is easily captured and cleaned by the reticuloendothelial system according to previous reports[20]. Therefore, the following vaccine studies were conducted using a mass ratio of 10:1 (CCD:RBD-HR). Meanwhile, the RBD-HR loading capacity of CCD was tested. When the mass ratio of CCD/RBD-HR was 10:1, it reached the maximal antigen-carrying rate at approximately 100% (Fig. 1h). CCD/RBD-HR (10:1) could form stable sphere nanoparticles (Fig. 1i), which could protect RBD-HR from enzymatic degradation in the nasal cavity during intranasal immunization[14]. Thus, with simple mixing, RBD-HR could bind to CCD to form stable nanoparticles as a result of electronic interactions.

### Immunogenicity and protective effects of intranasal CCD/RBD-HR vaccine

Animals were intranasally immunized at days 0, 14, and 28 with an immunization dose of 100 μg CCD plus 10 μg RBD-HR (per mouse) or 200 μg CCD plus 20 μg RBD-HR (per rabbit) (Fig. 2a). We observed that antigen-specific IgG titers in the serum were orders of magnitude higher when RBD-HR was administered with CCD than when RBD-HR was administered alone in both mice (Fig. 2b) and rabbits (Fig. 2c). Antibody subtype analysis revealed the production of IgG1, IgG2a, IgG2b, IgG2c, and IgG3 antibodies in mice, suggesting the induction of both T helper type 1 (Th1)- and Th2-biased immune responses by CCD/RBD-HR immunization (Supplementary Fig. 2a–e). The increased serum antibodies were highly functional, which blocked the specific binding between RBD and ACE2 receptors (Supplementary Fig. 3a, b) and neutralized the authentic viruses and pseudoviruses of SARS-CoV-2 (Fig. 2d, e). Moreover, the immune sera from mice immunized with CCD-adjuvanted RBD-HR broadly protected against infection with both the wild-type (WT) and common mutant strains of SARS-CoV-2, including Alpha, Beta, Delta, and Omicron variants (BA.1, BA.2, BA.2.12.1, BA.3, and BA.4/5) (Fig. 2e)[21]. We next analyzed the production of antigen-specific IgA antibodies. As expected, vigorous anti-RBD-HR IgA antibody responses were induced in the bronchoalveolar lavage (BAL) fluid and serum of mice vaccinated with CCD/RBD-HR but not in mice immunized with naked RBD-HR or PBS (Fig. 2f and Supplementary Fig. 2f). This indicated the establishment of strong mucosal immunity, which plays a significant role in the first-line defense against virus infection at local sites[8].

We next sought to evaluate the protective activity of the CCD-adjuvanted RBD-HR vaccine in WT female BALB/c mice that are sensitive to Omicron variant infection[22]. After vaccination with PBS, CCD, or CCD/RBD-HR, mice were challenged intranasally with $5 \times 10^5$ 50% tissue culture infective dose (TCID$_{50}$) of SARS-CoV-2 Omicron. Four days post-infection (dpi), mice were euthanized, and nasal turbinates,

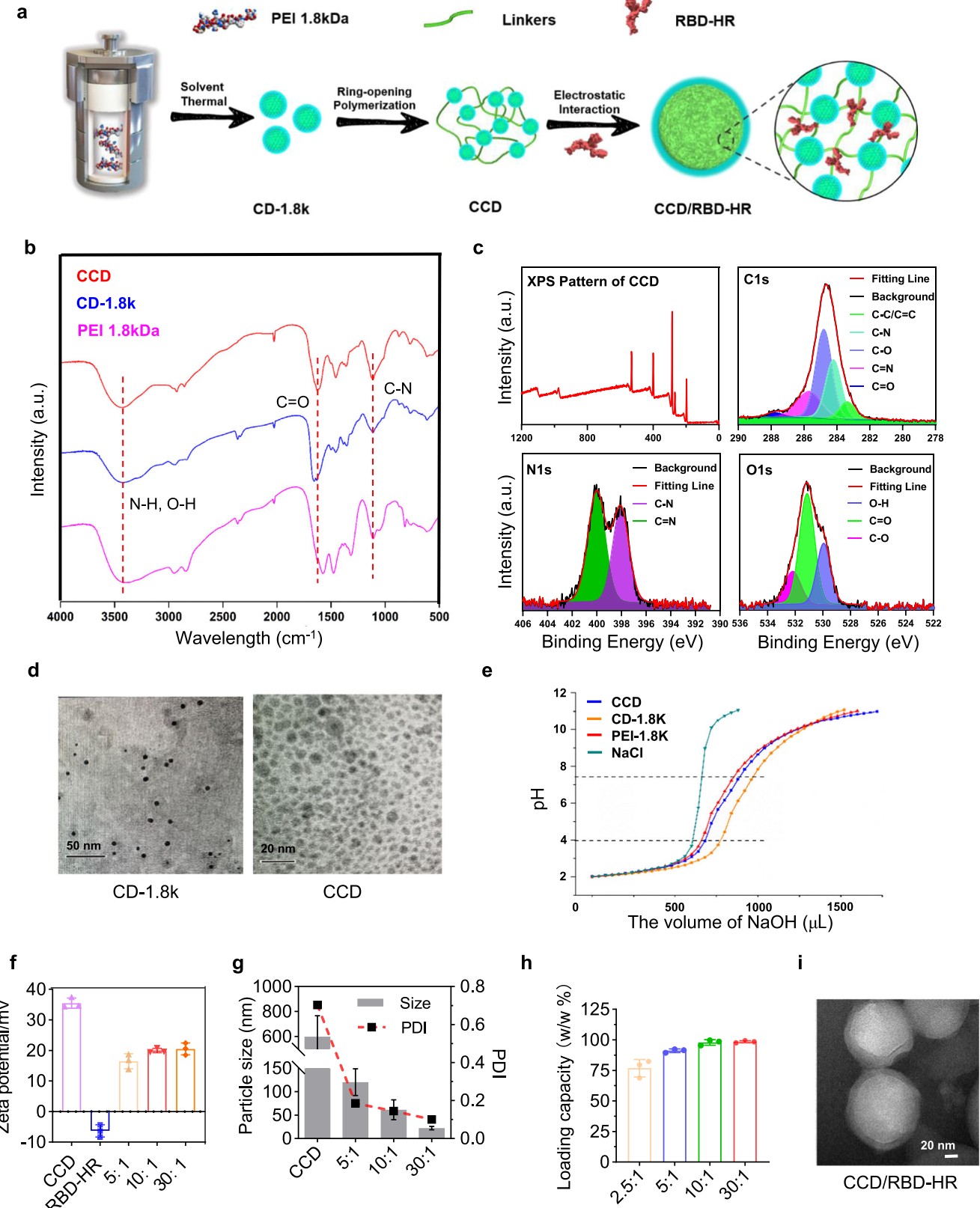

**Fig. 1 | Characterization of CCD and CCD/RBD-HR nanoparticles. a** The synthesis route of CCD. **b** The surface chemical stages of CCD are characterized by XPS. **c** The chemical structure of CCD, CD-1.8k, and PEI 1.8 kDa measured by FT-IR. **d** The morphology of CD-1.8k (left) and CCD (right) observed by TEM. **e** Acid–base titration profiles of CCD, CD-1.8k, PEI 1.8 kDa, and 150 mM NaCl solutions. **f**, **g** The zeta potentials (**f**) and particle sizes (**g**) of CCD/RBD-HR with different mass ratios. **h** The loading capacity of CCD/RBD-HR with different mass ratios (2 µg RBD-HR for each test). $n = 3$ independent experiments per group in (**f**–**h**). **i** The morphology of CCD/RBD-HR with a mass ratio of 10:1 was observed by TEM. Data were presented as mean values ± SEM in (**f**–**h**). Similar results for **d** and **i** were obtained in three independent experiments. Source data are provided as a Source Data file.

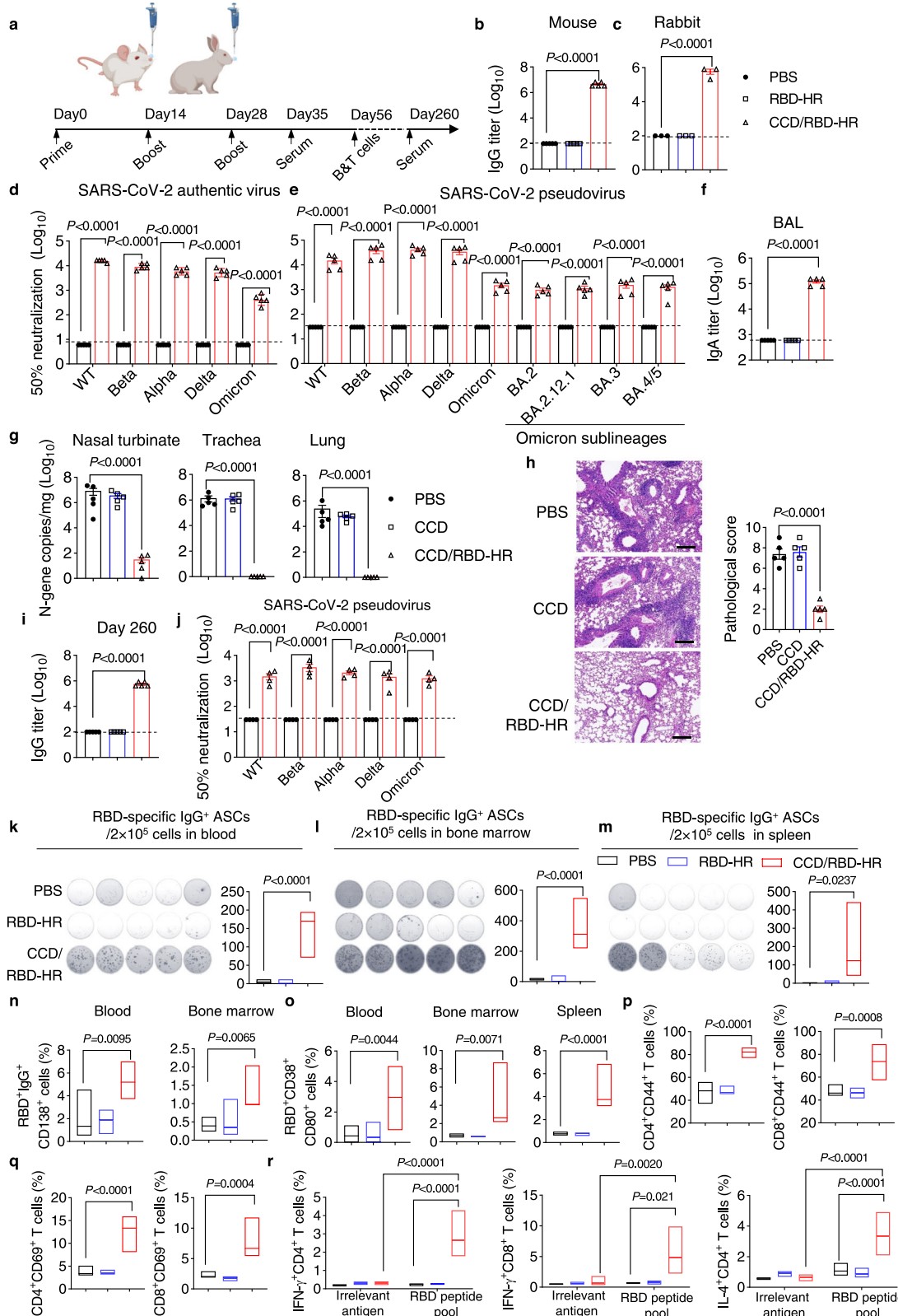

trachea, and lungs were harvested for the assessment of viral burden. Intranasal vaccination with CCD/RBD-HR conferred complete protection against Omicron infection both in the upper and lower respiratory tracts, as determined by almost undetectable viral RNA in nasal turbinates, trachea, and lungs (Fig. 2g). Histopathological evaluation revealed minimal or no infiltration in the lungs of mice vaccinated with

CCD/RBD-HR at 4 dpi compared to the apparent inflammation in mice immunized with PBS or CCD alone (Fig. 2h).

Given the ongoing pandemic status of SARS-CoV-2, the persistence of antibody responses is critical. A waning humoral immune response to the licensed BNT162b2 COVID-19 vaccine was observed six months after the second-dose immunization[23]. In our experiments,

**Fig. 2 | Immunogenicity and protective effects of intranasal CCD/RBD-HR vaccine. a** Immunization scheme. Mice/rabbits were immunized intranasally with PBS, RBD-HR, or CCD/RBD-HR on days 0, 14, and 28. **b, c** Serum anti-RBD-HR IgG titers in immunized mice (**b**) and rabbits (**c**) on day 35. **d, e** Neutralizing of authentic viruses (**d**) and pseudoviruses (**e**) of SARS-CoV-2 by the immune sera on day 35. **f** RBD-HR-specific IgA levels in BAL fluid on day 35. Immunized mice were challenged with $5 \times 10^5$ $TCID_{50}$ of Omicron viruses and sacrificed at 4 dpi. **g** Viral RNA levels in the nasal turbinates, trachea, and lungs at 4 dpi. **h** Histopathological changes (left) and pathological scores (right) of the lung tissues at 4 dpi. Scale bar: 100 µm. Similar results were obtained in five independent experiments. **i, j** Serum RBD-HR-specific IgG levels (**i**) and their function in neutralizing SARS-CoV-2 variant pseudoviruses (**j**) on day 260. On day 56, blood (**k**), bone marrow (**l**), and spleen (**m**) were harvested to assess SARS-CoV-2 RBD-specific IgG ASCs with ELISpot. Images (left) and quantification (right) of IgG spot-forming cells among lymphocytes were displayed. RBD-specific IgG-producing CD138+ plasma cells in the blood and bone marrow (**n**), memory B cells (RBD+CD38+CD80+) in the blood, bone marrow, and spleen (**o**) were assayed with FCM. **p, q** Antigen-experienced (CD44+) and activated (CD69+) CD4+/CD8+ T cells in the spleen were determined with FCM after re-stimulation with RBD protein peptide pools. **r** Splenic CD4+ and CD8+ T cells were assessed for IL-4 and IFN-γ expression with intracellular cytokine staining after restimulation with RBD protein peptide pools or irrelevant antigens. Data were displayed with floating bars in (**k–r**). The middle line indicates the median and the box shows the data range. Data were presented as mean values ± SEM. $n = 4–5$ mice in (**b, d–r**). $n = 3$ rabbits in (**c**). $P$ values were calculated with one-way ANOVA followed by Dunnett's multiple comparisons test in (**b, c, f–i, k–q**). Two-way ANOVA followed by Sidak's multiple comparisons test in (**d, e, and j**). Two-way ANOVA followed by Tukey's multiple comparisons test in (**r**). Source data are provided as a Source Data file.

following RBD-HR immunization with CCD, the titers of RBD-HR-specific IgG antibodies in the serum remained relatively high 260 days after the prime immunization, which was comparable to that on day 35 (Fig. 2i). Serum collected on day 260 from mice immunized with CCD/RBD-HR was active in neutralizing the infection of both ancestral and mutant pseudoviruses of SARS-CoV-2 (Fig. 2j).

Ideal vaccine immunity consists of humoral and cellular immunity[24]. Thus, we next investigated the activation of antigen-specific T-cell and B-cell responses both locally and systemically after vaccination. Mice were immunized as mentioned before. Four weeks after the third immunization, lymphocytes from the blood, bone marrow, and spleen were harvested and analyzed by ELISpot and flow cytometry (FCM). ELISpot analysis with RBD protein indicated that CCD/RBD-HR immunization induced high levels of RBD-specific IgG antibody-secreting cells (ASCs) in blood, bone marrow, and spleen, whilst the control RBD-HR did not (Fig. 2k–m). A fluorescently labeled RBD protein was produced to gate out RBD-binding B cells (Supplementary Fig. 4a). Similarly, we found that CCD/RBD-HR immunization resulted in an elevation in the frequency of RBD-specific IgG-producing CD138+ plasma cells (PCs) in the circulation and bone marrow with FCM (Fig. 2n, Supplementary Fig. 4b–d), further supporting the activation of strong B-cell responses. Memory B cells (MBCs, CD38+CD80+) can enhance rapid recall responses upon secondary exposure[25]. As a mucosal adjuvant, CCD substantially promoted the expansion of RBD-specific MBCs in the blood, bone marrow, and spleen (Fig. 2o, Supplementary Fig. 4e), accounting for the sustainability of humoral immunity. Next, splenic CD3+ T cells from the immunized mice were isolated and stimulated ex vivo with overlapping peptide pools for RBD. Peptides for the model antigen OVA ($OVA_{257-264}$ and $OVA_{323-339}$) were used as irrelevant antigen control. After restimulation with RBD peptides rather than OVA peptides, antigen-experienced (CD44+) and activated (CD69+) CD4 T+ and CD8 T+ cells in the spleen were significantly elevated in mice vaccinated with CCD/RBD-HR (Fig. 2p, q, Supplementary Fig. 5a, b). The production of interferon-γ (IFN-γ) and interleukin-4 (IL-4) was determined by intracellular cytokine staining. We observed a higher proportion of splenic CD4 T+ and/or CD8 T+ cells expressing IL-4 and IFN-γ in mice inoculated with CCD/RBD-HR compared with pure RBD-HR immunization (Fig. 2r, Supplementary Fig. 5c), demonstrating the activation of antigen-specific T-cell immunity systemically.

## Vaccine-induced local T- and B-cell responses

To assess the effect of intranasal immunization on mucosal immunity, vaccinated mice were sacrificed 4 weeks after the last vaccination to determine the immune responses in the lungs, BAL fluid, and draining cervical lymph nodes (CLNs). A substantial increase in the frequencies of T follicular helper (Tfh, CD4+CXCR5+PD-1+) and antigen-specific germinal center B (GCB, GL7+CD95+RBD+) cells in the CLN was observed in the CCD/RBD-HR group compared with immunization with RBD-HR alone (Fig. 3a, Supplementary Fig. 6a, b), which was critical for the generation of strong and long-term humoral immunity against SARS-CoV-2[26,27]. Then we also tested antigen-specific T-cell responses in the lungs after immunization. Lung tissues were harvested 4 weeks post-boosting and T cells were evaluated with FCM. Restimulation with a pool of RBD peptides induced a remarkable elevation in the frequency of IFN-γ-generating CD8+ and CD4 T+ cells in the lungs of mice receiving CCD/RBD-HR vaccine (Fig. 3b, c, Supplementary Fig. 7). In contrast, irrelevant antigen restimulation showed minimal effects on IFN-γ producing, indicating the induction of antigen-specific T cell responses. Given that RBD-HR-specific major histocompatibility complex-I (MHC I) and MHC II multimers were not available at the time of this study, we used the model antigen OVA to further verify the generation of antigen-specific T-cell responses. The induction of OVA-specific T-cell responses in the lungs was evaluated by H-2K$^b$-restricted $OVA_{257-264}$ (SIINFEKL) and I-A$^d$-restricted $OVA_{323-339}$ tetramer staining. As shown in Supplementary Fig. 8a, b, the addition of CCD adjuvant induced a sharp increase in the production of tetramer-positive CD4+ and CD8+ T cells.

Accumulating evidence indicates the critical role of mucosal tissue-resident memory T cells ($T_{RMS}$) in host defense against viral infection[28]. Thus, we next explored whether CCD could promote the formation of respiratory mucosal $T_{RM}$ cells. Mice vaccinated with CCD/RBD-HR exhibited a marked enhancement in the percentages and numbers of antigen-experienced CD4+ and CD8+ T cells in the lungs (Supplementary Fig. 9, Supplementary Fig. 10a–d) four weeks after the last immunization. The increased antigen-experienced CD4+ and CD8+ T cells upon CCD exposure expressed substantially elevated levels of CD69, CD103, or both, which are typical markers for vaccine-induced $T_{RM}$s (Fig. 3d, Supplementary Fig. 10e–g)[29,30]. These data suggested the capacity of CCD to guide the formation of lung $T_{RM}$ cells after vaccination, including both CD4+ and CD8+ $T_{RM}$ cells. As previously reported[30], we speculated that the formation of lung $T_{RM}$ cells requires the migration of T cells from the draining lymph nodes, at least after the prime immunization, since we found that CD4+CD69+ and CD8+CD69+ T cells in the draining lymph nodes were conversely decreased in mice immunized with CCD/RBD-HR (Supplementary Fig. 11a, b). Seven days after the third immunization, we discovered that CCD/RBD-HR vaccination elicited an approximately threefold increase in the fraction of lung-resident (CD103+) DCs that expressed high levels of the activation markers CD86 and MHC II in immunized mice compared with pure RBD-HR (Fig. 3e).

We next evaluated $T_{RM}$ cells within the airways (BAL) 4 weeks after the final immunization. t-SNE maps were established based on pooled CD3+ T cells from BAL fluid and heatmaps were produced to overlay the expression intensities of CD4, CD8, CD44, CD69, and CD103[6]. Intranasal delivery of CCD/RBD-HR contributed to a drastic elevation in airway CD4+, CD8+, and antigen-experienced CD4 or CD8 T+ cells (Fig. 3f). Meanwhile, we observed the formation of several large clusters of CD4+ and CD8+ T cells with $T_{RM}$ phenotypes after intranasal vaccination with CCD/RBD-HR (Fig. 3f). We also determined

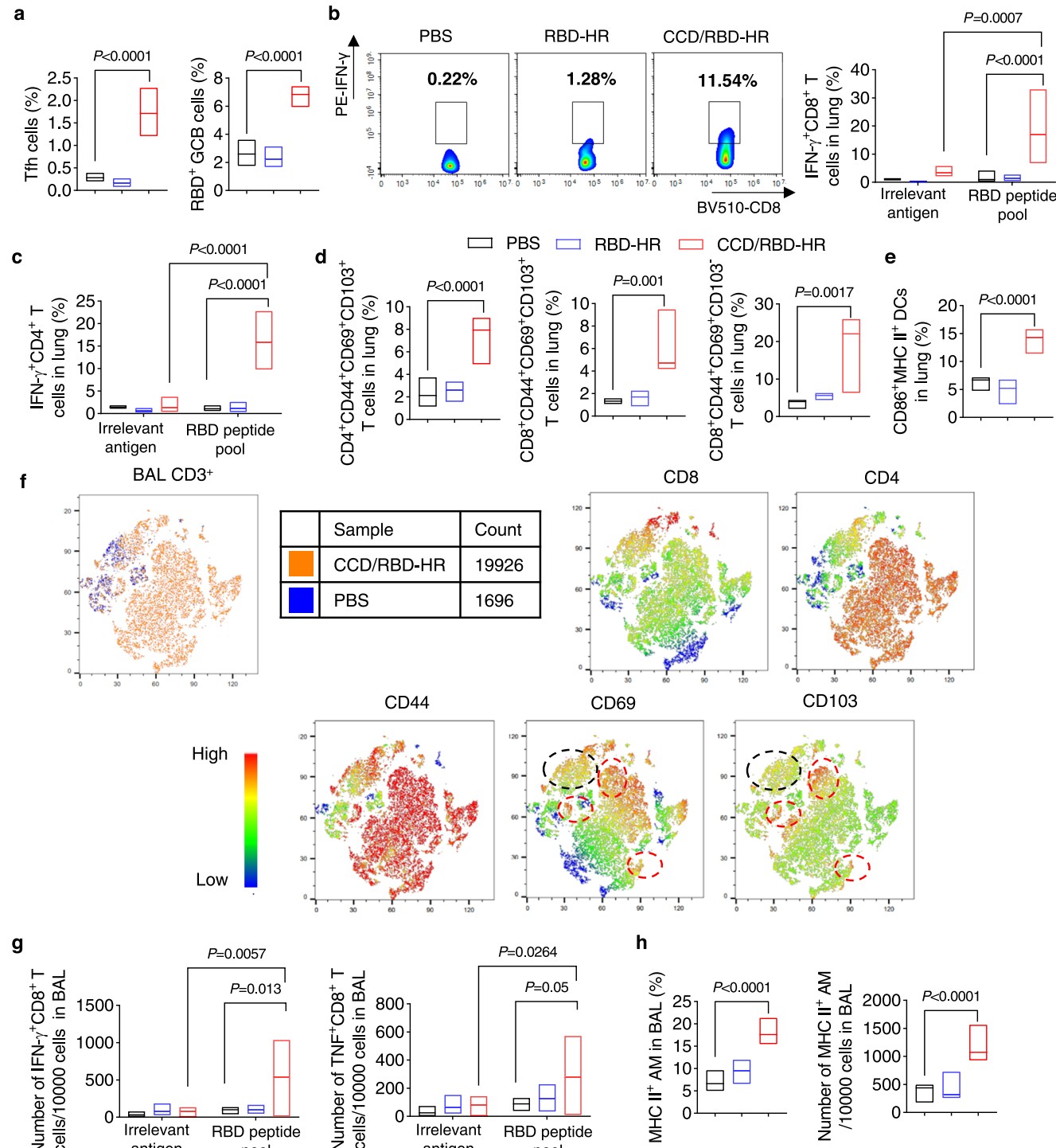

**Fig. 3 | CCD/RBD-HR immunization induces local T-cell responses.**
**a** The percentages of Tfh (CD4⁺CXCR5⁺PD-1⁺) and RBD-specific GCB (B220⁺GL7⁺CD95⁺RBD⁺) cells in the CLN were analyzed with FCM at 28 days after the last immunization. **b**, **c** Lung CD8⁺ and CD4⁺ T cells were evaluated for IFN-γ generation by FCM upon restimulation with RBD peptide pools or irrelevant antigens. **b** Representative FCM plots (left) and quantification (right) of IFN-γ-expressing lung CD8⁺ T cells. **d** The proportion of antigen-experienced (CD44⁺) CD4⁺ and CD8⁺ T cells expressing CD69 and CD103 in lung tissues. **e** The percentage of CD103⁺ DCs coexpressing CD86 and MHC II in the lungs. **f** Upper left: t-SNE maps were created by concentrating CD3⁺ T cells from the BAL fluid of the vaccinated mice. Analysis was carried out with default FlowJo V.10 software. Upper right and bottom: heat-map projections of CD4, CD8, CD44, CD69, and CD103 expression on t-SNE maps

from three independent mice ($n = 3$). Red and black hashed circles are indicators of antigen-experienced CD4⁺ and CD8⁺ T$_{RM}$ cells in the BAL fluid, respectively. **g** Absolute numbers of CD8⁺ T cells producing IFN-γ (left) and TNF-α (right) in the BAL after ex vivo restimulation with RBD peptide pools or irrelevant antigen at 4 weeks after the last immunization. **h** Quantification of the percentages (left) and the numbers (right) of MHC II⁺ AM in the BAL fluid. Data were displayed with floating bars in (**a**–**e**, **g**, and **h**). The middle line indicates the median and the box shows the data range. Data were presented as mean values ± SEM. $n = 4$–6 mice in each group in (**a**–**e**, **g**, and **h**). $P$ values in **a**, **d**, **e**, **h** were calculated with One-way ANOVA followed by Dunnett's multiple comparisons tests. $P$ values in **b**, **c**, and **g** were calculated with two-way ANOVA followed by Tukey's multiple comparisons tests. Source data are provided as a Source Data file.

antigen-specific T cells with intracellular IFN-γ and TNF-α expression in mononuclear cells of the BAL with FCM upon ex vivo restimulation with peptide pools for RBD or OVA peptides control. As expected, CD8[+] T cells from CCD/RBD-HR-immunized mice responded to peptide pools for RBD four weeks after the last boosting (Fig. 3g). To assess the activation of innate immunity, BAL fluids were collected seven days after the third immunization. The results indicated that aside from activating strong adaptive immune responses, CCD/RBD-HR immunization also induced a robust trained alveolar macrophage (AM) response within the airway (Fig. 3h), which is characterized by co-expression of MHC II and CD11b and may be vital in the early innate immune control of COVID-19[6,31].

## CCD/RBD-HR vaccine promotes dendritic cell activation

To investigate the innate immune response, immune cell recruitment in the CLN was determined with FCM 72 h after vaccination (Supplementary Fig. 12)[32]. As shown in Fig. 4a, CCD/RBD-HR induced higher recruitment of infiltrating monocytes, CD11b[−] DCs, and CD11b[low] DCs than pure RBD-HR. Other immune cells, including macrophages, eosinophils, neutrophils, and myeloid DCs (mDCs), did not change in the CLN. Next, we explored the impact of CCD/RBD-HR on the most focused APCs, DCs, which are strong in presenting antigens to naive T cells[33]. We labeled RBD-HR with 488 fluorescent dyes to visualize or quantify its interaction with different cells. The uptake of antigens by bone marrow-derived DCs (BMDCs) was assessed with FCM using the fluorescently labeled RBD-HR. We found that naked RBD-HR incubation showed modest antigen incorporation, while RBD-HR mixed with CCD displayed a sharp increase in antigen uptake by BMDCs (Fig. 4b). We also observed that immunization with CCD/RBD-HR, rather than RBD-HR alone, promoted the maturation of DCs in the draining CLN in mice, as characterized by the elevation in the expression of maturation markers CD40, CD80, CD86 and MHC II (Fig. 4c). Incubation of BMDCs with CCD/RBD-HR for 24 h resulted in a remarkable elevation in the expression of CD40 and CD80 in vitro as well (Fig. 4d)[34]. A magnetic Luminex assay revealed that CCD/RBD-HR treatment enhanced the production of multiple proinflammatory cytokines from BMDCs, including IL-1β, IL-6, MCP-1, etc. (Fig. 4e). Efficient presentation of antigens by APCs is critical for the initiation of immune responses[35]. We next labeled splenic T cells with CFSE fluorescence and incubated them with BMDCs pre-stimulated with PBS, RBD-HR, or CCD/RBD-HR. BMDCs treated with CCD/RBD-HR, but not RBD-HR, induced a significantly elevated degree of proliferation and activation (CD69[+], CD44[+]) in both CD4[+] and CD8[+] T cells (Fig. 4f–h), along with an enhanced secretion of IL-4 and IFN-γ in the supernatants (Fig. 4i) during the coculture process. Therefore, concomitant DC activation may be involved in the mechanism of CCD's effect on enhanced immunity after intranasal immunization.

## CCD/RBD-HR vaccine targets nasal epithelial cells as potent APCs at mucosal

In addition to professional APCs, recent studies indicated that antigen-presenting epithelial cells can present antigens at local sites, which is essential for the regulation of barrier tissue immunity, including the recruitment and maintenance of T_RM cells[36–38]. Using the fluorescently labeled antigens, we observed that intranasally delivered CCD/RBD-HR, but not naked RBD-HR, could still be universally attached to the nasal epithelium layer 24 h after intranasal immunization. (Fig. 5a). Immunohistochemical staining revealed that 30 min after immunization, the density of RBD-HR in the nasal mucosa was much greater when it was incorporated into the CCD (Supplementary Fig. 13). 24 h later, naked RBD-HR was almost completely removed from the nasal cavity, whereas increased RBD-HR binding and penetration between NECs was observed when it was administered with CCD. We next determined whether NECs can present the intranasally delivered

vaccine, which can be discriminated based on their expression of epithelial cell adhesion molecule (EpCAM)[36].

Mice were immunized intranasally with the labeled RBD-HR with or without CCD. 24 h later, the nasal mucosa was harvested, and the uptake of antigens into EpCAM[+] NECs was examined with FCM. Strikingly, RBD-HR can only be taken up by NECs when it is mixed with CCD (Fig. 5b). The same result was observed when NECs from nasal mucosa were isolated and treated with RBD-HR or CCD/RBD-HR ex vivo (Fig. 5b). We found that CCD could produce a strong proton sponge to promote lysosomal escape (Supplementary Fig. 14), which contributed to its good buffer capacity, as proven in Fig. 1e. We also found that CCD/RBD-HR treatment significantly promoted the maturation of NECs by upregulating the expression of MHC II and costimulatory molecules (CD40, CD80, CD86) both in vivo (Fig. 5c) and in vitro with a mouse NEC cell line (Fig. 5d). Confocal microscopy displayed more RBD-HR antigens colocalized with MHC II in primary NEC treated with CCD/RBD-HR, suggesting more effective presentation of antigens upon CCD stimulation (Fig. 5e). Enhanced production of IL-6 in the supernatant was observed when NEC cell lines were stimulated with CCD/RBD-HR (Fig. 5f). Similar to what was observed in BMDCs, primary NECs pre-stimulated with CCD/RBD-HR promoted the proliferation (Fig. 6d) and increased the fractions of activated and antigen-experienced CD4[+] T cells (Fig. 5g), which was completely reversed by a specific neutralizing antibody against MHC II (Fig. 5h). We also discovered a significant increase in the proportion of proliferated CD8[+] T cells upon coculture with the NEC cell line stimulated with CCD/RBD-HR (Fig. 6e). Collectively, these data strongly supported the role of NECs as potent APCs that can present antigens to and activate T cells at local sites. Considering that the number of NECs far exceeds that of professional APCs in the nasal mucosa, NECs may play an indispensable role in the induction of antigen-specific immunity after mucosal vaccination with CCD/RBD-HR.

## CCD promotes antigen binding to NECs via sialic acids

We next investigated how CCD/RBD-HR was taken up by NECs using His-tagged RBD-HR. A markedly increased proportion of isolated primary NECs and NEC cell lines were positive for RBD-HR binding when treated with CCD/RBD-HR for 30 min relative to pure RBD-HR treatment, as determined with FCM (Fig. 6a). We then explored how CCD promotes antigen binding to epithelial cells. Neu5Ac are ubiquitous carbohydrates located on the end of glycoproteins and glycolipids on the surface of eukaryotic cells, which are highly responsible for the negative charge of the cell surface and have been reported to mediate viral attachment and entry into host cells[39–41]. Considering the cationic property of CCD, we hypothesized that CCD may promote antigen binding by interfering with Neu5Ac. Therefore, we pretreated epithelial cells with neuraminidase (NA) to remove cell surface Neu5Ac or incubated CCD with Neu5Ac in advance to prevent its interaction with Neu5Ac on cells. As displayed in Fig. 6a, both NA pretreatment and Neu5Ac incubation severely hampered antigen binding to primary NECs and NEC lines. The same trend was further confirmed using fluorescence microscopy in NEC lines (Fig. 6b). However, NAs or Neu5Ac treatment had no significant effect on the binding of RBD-HR to BMDCs, indicating that CCD-promoted antigen binding to BMDCs is reliant on other mechanisms instead of Neu5Ac (Fig. 6a). Reduced binding of RBD-HR to nasal mucosa was also observed in vivo when mice were pretreated intranasally with NAs before CCD/RBD-HR immunization or treated with Neu5Ac-containing CCD/RBD-HR compared to only CCD/RBD-HR administration (Fig. 6c). We presumed that reduced antigen binding may result in less antigen incorporation into the antigen-presenting NECs and attenuated immune responses. In line with our expectations, in the presence of NAs or Neu5Ac, the CCD/RBD-HR-stimulated NEC lines exhibited a reduced ability to present antigens to both CD4[+] and CD8[+] T cells, as implied by a reduced proportion of proliferated, activated, and antigen-experienced T cells

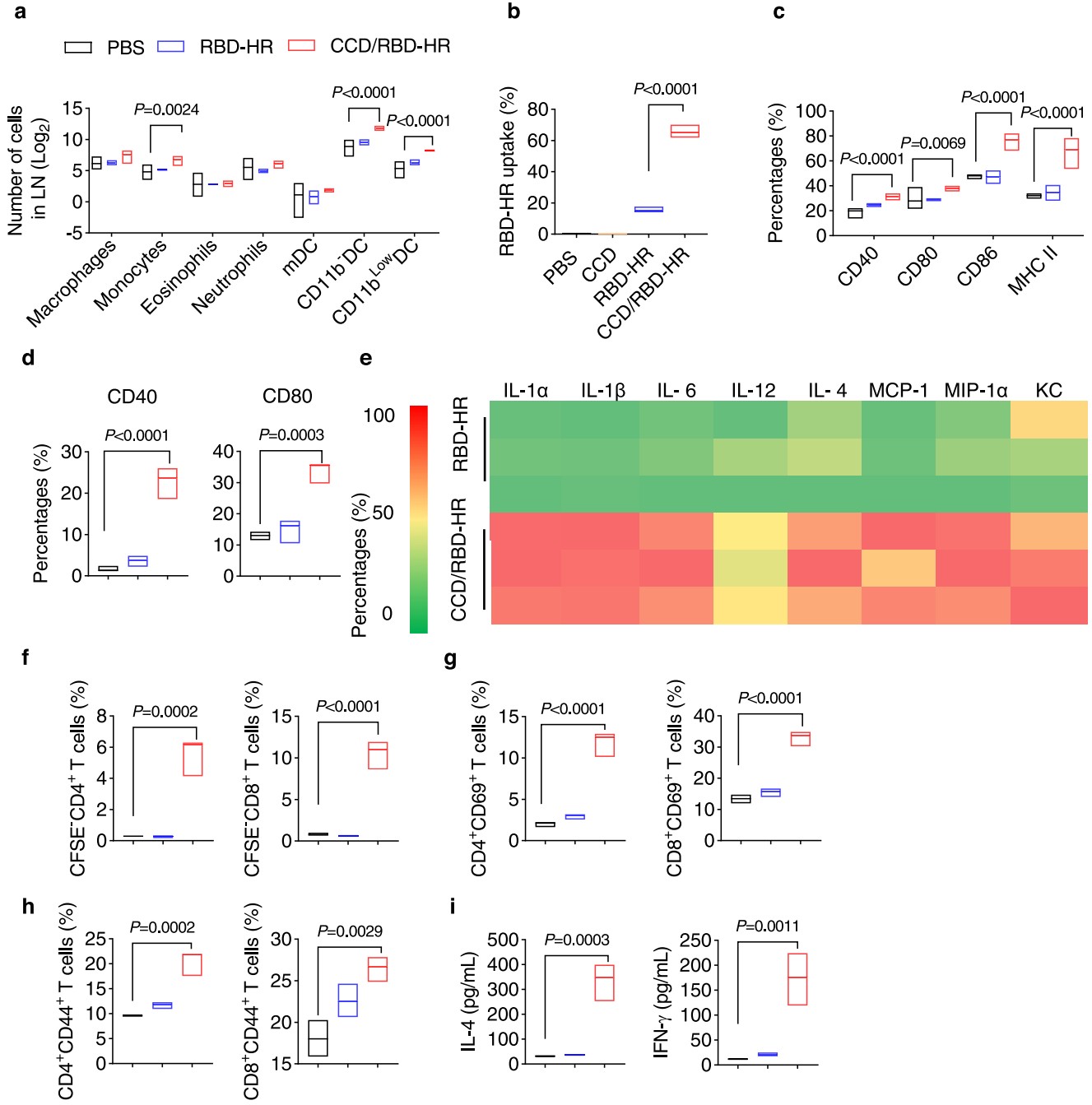

**Fig. 4 | The CCD/RBD-HR vaccine targets dendritic cells. a** Immune cell recruitment in the CLN 72 h after intranasal immunization with PBS, RBD-HR, or CCD/RBD-HR was determined with FCM. **b** RBD-HR uptake after incubating mouse BMDCs with or without CCD for 30 min. **c** Maturation of DCs in the CLN 72 h after intranasal immunization with PBS, RBD-HR, or CCD/RBD-HR as determined by the expression of CD40, CD80, CD86, and MHC II. **d, e** BMDCs were treated with PBS, RBD-HR, or CCD/RBD-HR for 24 h. **d** Maturation of BMDCs was determined by the expression of CD40 and CD80 with FCM. **e** Magnetic Luminex assay test of cyto-kines in the supernatant. The proliferation (**f**) and fractions of CD69+ (**g**) and CD44+

(**h**) of both CD4+ and CD8+ T cells were determined with FCM after T cells were co-incubated with pre-stimulated DCs for 48 h. (**i**) Cytokines (IL-4 and IFN-γ) in the supernatant were determined by ELISA. Data were displayed with floating bars in (**a–d**) and (**f–i**). The middle line indicates the median and the box shows the data range. Data were presented as mean values ± SEM. $n = 5$ mice per group in **a** and **c**. Three independent experiments ($n = 3$) were performed in each group in (**b**, **d–i**). $P$ values were calculated with One-way ANOVA followed by Dunnett's multiple comparisons tests. Source data are provided as a Source Data file.

(Fig. 6d, e). In addition, the geometric mean titers of serum IgG antibodies in CCD/RBD-HR-immunized mice were reduced when mice were pretreated with NAs via the intranasal route (Fig. 6f). The same results were observed when Neu5Ac was added to the CCD/RBD-HR mixture before immunization. Therefore, the induction of the humoral immune response by CCD/RBD-HR is dependent on the early binding of antigens to NECs via Neu5Ac.

Then, we explored the uptake mechanism with the fluorescently labeled RBD-HR. The cellular uptake rate of RBD-HR decreased to below 10% when NEC cell lines were incubated with CCD/RBD-HR at 4 °C for 4 h, suggesting an energy-dependent mechanism for cell entry (Fig. 6g). Then, NEC cell lines were pretreated with various inhibitors of different uptake pathways before incubation with CCD/RBD-HR, among which cytochalasin D treatment resulted in a more

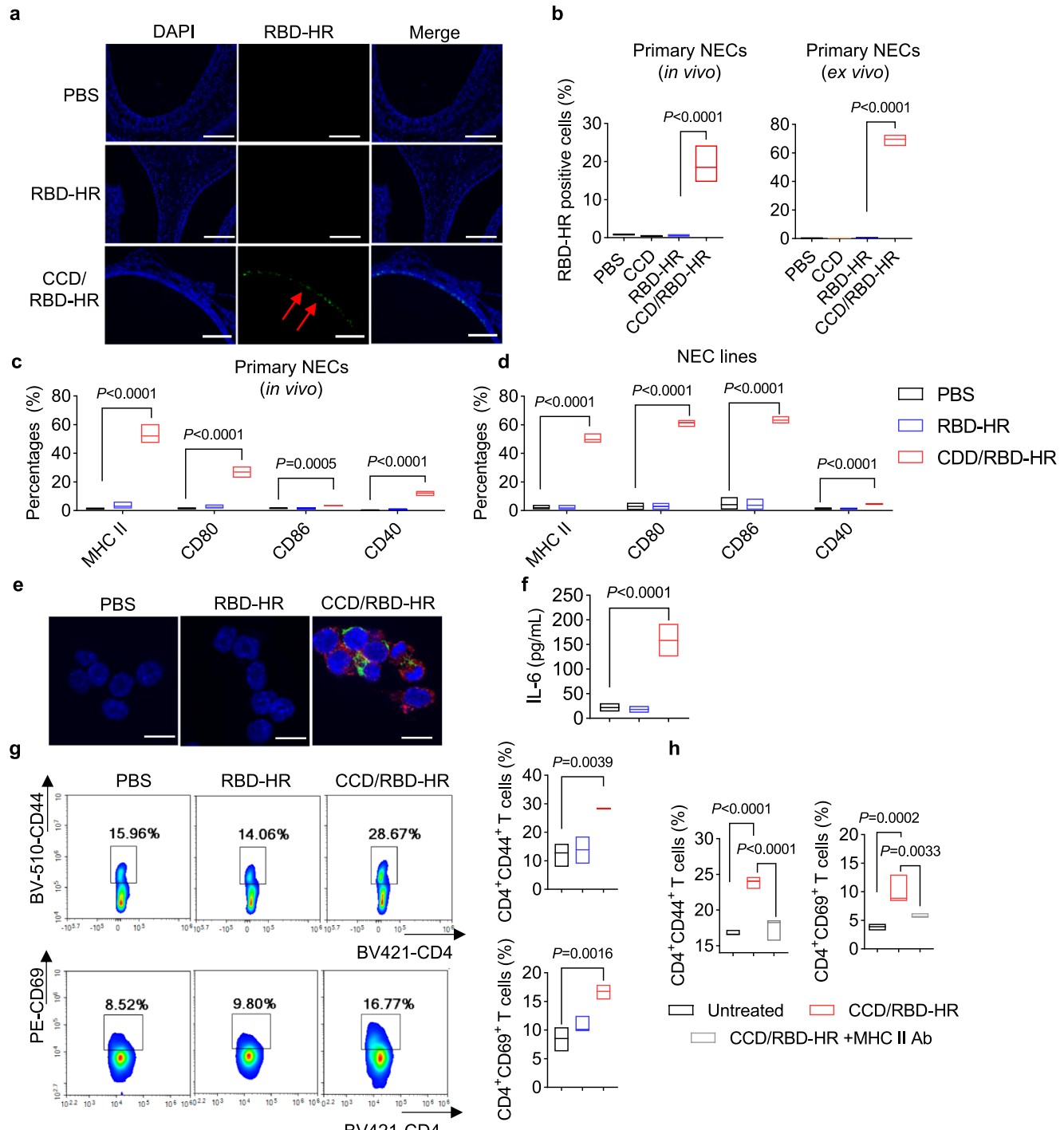

**Fig. 5 | The CCD/RBD-HR vaccine targets nasal epithelial cells. a** Representative immunofluorescent images of nasal mucosa harvested from BALB/c mice (*n* = 3) immunized with PBS or RBD-HR (green) with or without CCD intranasally 24 h before. Blue: DAPI. Scale bars: 100 μm. **b** Left: BALB/c mice (*n* = 3) were immunized with PBS or fluorescently labeled RBD-HR with or without CCD intranasally. 24 h later, the nasal mucosa was harvested, and the uptake of RBD-HR by primary NECs (EpCAM⁺) was assessed with FCM. Right: RBD-HR uptake by isolated primary NECs ex vivo after incubation with RBD-HR with or without CCD for 30 min. **c** Maturation of mouse NECs 24 h after immunization as determined by the expression of CD40, CD80, CD86, and MHC II using FCM. *n* = 3 mice per group. **d** A mouse NEC cell line was stimulated with PBS, RBD-HR, and CCD/RBD-HR for 24 h, and the expression of maturation markers was examined. **e** Mouse NECs were isolated and incubated with PBS or fluorescently labeled RBD-HR (green) with or without CCD for 24 h. Confocal

microscopy was used to image the localization of RBD-HR (green) and MHC II (red) on primary NECs. Blue: DAPI. Scale bars: 10 μm. Similar results were obtained in three independent experiments. **f** Concentration of IL-6 in the supernatant of the stimulated NEC cell line at 24 h. **g**, **h** Mouse primary NECs were isolated and stimulated with PBS, RBD-HR, or CCD/RBD-HR in the presence or absence of anti-MHC II antibodies for 24 h, followed by co-culture with splenic T cells. 48 h later, fractions of CD4⁺CD44⁺ and CD4⁺CD69⁺ T cells were assessed with FCM. Data were displayed with floating bars in (**b**–**d**) and (**f**–**h**). The middle line indicates the median and the box shows the data range. Data were presented as mean values ± SEM of three (*n* = 3) in **b**, **g**, and four (*n* = 4) independent experiments in (**d**, **f**, **h**). *P* values were calculated with One-way ANOVA followed by Dunnett's multiple comparisons tests. Source data are provided as a Source Data file.

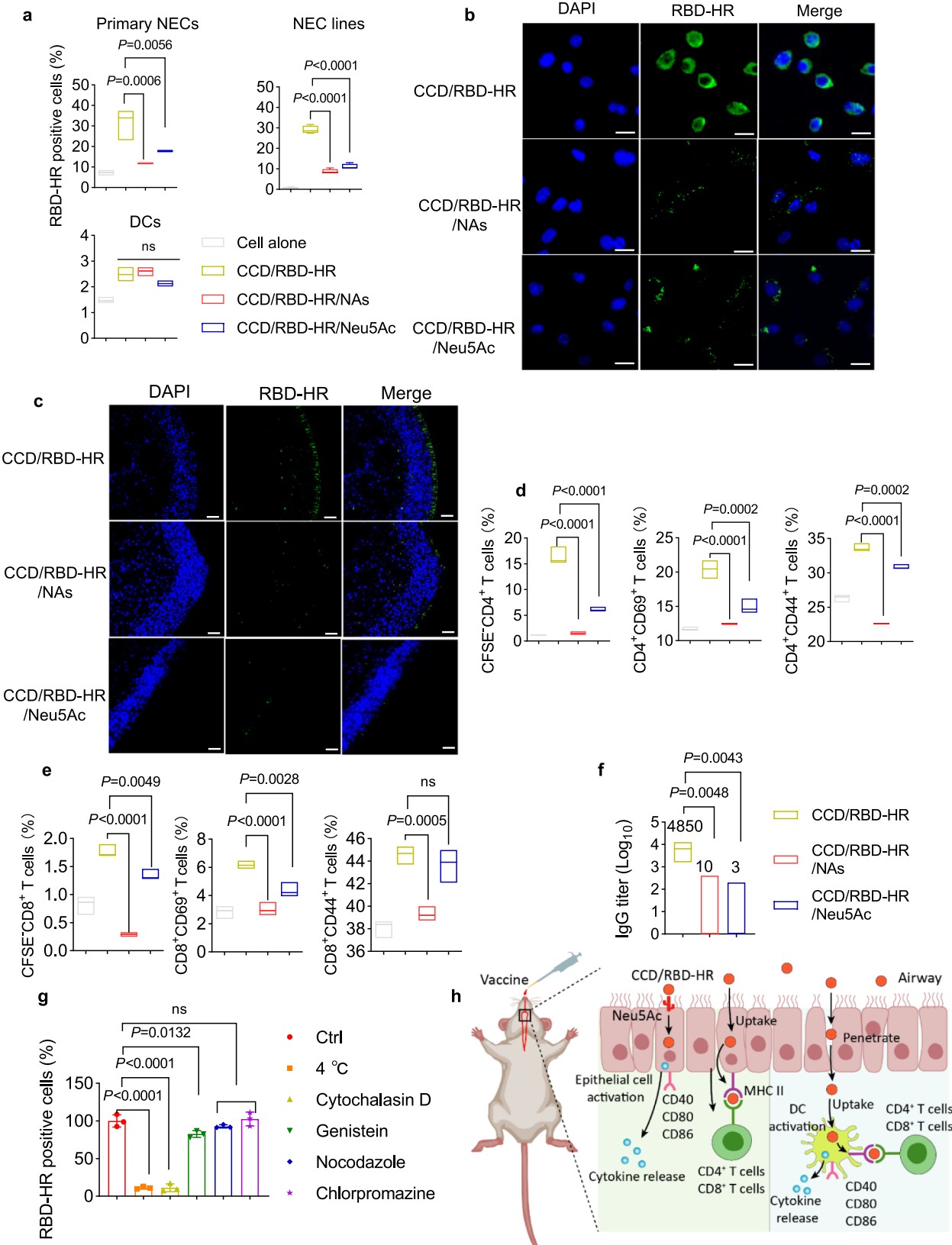

than 80% reduction in antigen uptake, suggesting that the main internalization of CCD/RBD-HR into NECs was micropinocytosis-dependent (Fig. 6g)[42]. Thus, CCD may promote antigen binding to NECs via Neu5Ac and further promote antigen uptake through energy-dependent micropinocytosis.

## Safety evaluation

To assess the safety of the intranasal CCD/RBD-HR vaccine, a complete blood count (Fig. 7a) and serum chemistry panel (Fig. 7b) analysis were performed one week after the third immunization. No statistically significant difference was observed between the groups. No histologically

**Fig. 6 | CCD promotes antigen binding to NECs via sialic acids. a** FCM assessment of the percentages of His-tagged RBD-HR binding to primary NECs ($n = 3$, up left), NEC lines ($n = 4$, top right), and BMDCs ($n = 3$, bottom) in the presence of CCD when cells were pretreated with NAs or treated with Neu5Ac. Untreated cells or cells treated with CCD and His-tagged RBD-HR were used as the negative and positive controls, respectively. NAs: neuraminidase; Neu5Ac: sialic acid. **b** Fluorescence microscopy of NEC cell lines after incubation with His-tagged RBD-HR (green) and CCD in the presence or absence of NAs/Neu5Ac. Blue: DAPI. Scale bars: 25 μm. Similar results were obtained in three independent experiments. **c** BALB/c mice ($n = 3$) were immunized intranasally with CCD and His-tagged RBD-HR in the presence or absence of Neu5Ac. A group of mice was pretreated with NAs intranasally before receiving CCD/RBD-HR. The binding of His-tagged RBD-HR (green) to the nasal mucosa was evaluated with an immunofluorescence microscope. Blue: DAPI. Scale bars: 25 μm. **d**, **e** Mouse NEC lines were treated with CCD and RBD-HR with or without NAs/Neu5Ac for 24 h before coculturing with CFSE-labeled splenic T cells. The proliferation and fractions of CD69+ and CD44+ of both CD4+ (**d**) and CD8+ (**e**) T cells were determined with FCM after 48 h-coculture. **f** BALB/c mice ($n = 5$/group) were intranasally vaccinated with CCD/RBD-HR with or without NA pretreatment/Neu5Ac treatment. On day 14 postvaccination, anti-RBD-HR IgG antibodies in the sera were detected. **g** Uptake of fluorescently labeled RBD-HR by NEC lines in the presence of CCD when cells were treated at 4 °C or with cytochalasin D, nocodazole, genistein, and chlorpromazine. **h** Schematic illustration of the vaccine mechanisms. Data were displayed with floating bars in (**a**) and (**d**–**f**). The middle line indicates the median and the box shows the data range. Data were presented as mean values ± SEM of three independent experiments ($n = 3$) in (**d**, **e**, **g**). $P$ values were calculated with One-way ANOVA followed by Dunnett's multiple comparisons tests. Source data are provided as a Source Data file.

significant changes were observed in the lungs, spleen, heart, kidney, liver, or nasal mucosa of the immunized mice (Fig. 7c).

## Discussion

SARS-CoV-2 vaccines against emerging VOCs are urgently needed. What is equally warranted is SARS-CoV-2 vaccines that can simultaneously induce both local mucosal immunity at the respiratory tracts as well as systemic immunity, which are essential to prevent viral infection and transmission. To date, over 100 mucosal vaccines are under development against COVID-19 globally, with around 20 candidates reaching clinical evaluation and 4 vaccines approved for emergency use[43,44]. These needle-free vaccines are delivered via nose or mouth as an aerosolized mist or drops. Compared with other vaccine platforms, subunit protein vaccines possess the advantages of easy mass production, safety, and convenient transportation without the requirement for ultracold storage. Our team recently developed a self-assembling trimeric subunit vaccine RBD-HR which induced broadly protective immunity against omicron-included variants of SARS-CoV-2[5]. However, as mucosal vaccines, naked protein antigens showed poor immunogenicity since they can be cleared swiftly by mucociliary removal, necessitating the development of mucosal adjuvants[8]. In this study, we developed CCDs that showed excellent activity in shaping the formation of anti-viral immunity induced by RBD-HR after intranasal administration.

Three-dose immunization with CCD/RBD-HR efficiently induced strong and antigen-specific IgG antibodies in mice and rabbits with neutralization activity against WT and various VOCs of Alpha, Beta, Delta, and Omicron lineages. Once stimulated, such neutralizing antibody responses can last for a long period of time after vaccination, avoiding the requirements for repeated dosing. Meanwhile, strong IgA titers were detected both in the airway and circulation, indicating the establishment of mucosal immunity. On the contrary, no detectable antibody responses were observed in mice immunized with pure RBD-HR. Furthermore, the intranasal CCD/RBD-HR vaccine also provided protection from viral loads both in the upper and lower respiratory tracts in highly susceptible BALB/c mice after the challenge with the currently dominating Omicron variant.

The underlying mechanism of vaccine-induced immune responses was explored in mice. Defective GCB and Tfh cell responses were related to dysregulated and limited durability of humoral responses after SARS-CoV-2 infection[45]. Here we found that at four weeks after the third immunization with CCD/RBD-HR, robust antigen-specific GCB and Tfh responses were evoked in the draining lymph nodes (Fig. 3a). In consistent, a high proportion of RBD-specific MBC in the blood, spleen, and bone marrow were detected upon CCD/RBD-HR exposure (Fig. 2p). These results agreed with the durability of serum IgG responses (Fig. 2i, j) and proved the establishment of humoral immune memory to our intranasal vaccine[46–48]. We further determined T-cell immunity induced by the intranasally delivered CCD/RBD-HR. CCD/RBD-HR induced antigen-specific CD4+ and CD8+ T cell

responses, including a high proportion of IFN-γ-secreting cells upon ex vivo restimulation with peptides pools for RBD. These results were consistent with reports from our previous study[8]. Intranasal administration of CCD/RBD-HR also induced robust mucosal cellular immunity. Large proportions and numbers of antigen-specific CD4+ and CD8+ T cell responses were observed in the lungs and BAL fluid in mice immunized with CCD/RBD-HR, instead of naked RBD-HR, including CD103+ and/or CD69+ $T_{RM}$ cells (Fig. 3b–h, Supplementary Fig. 10). Respiratory $T_{RM}$ cells are essential components of first-line defense against viral infection at local sites without recirculating[49–51]. Previous studies with adenovirus-vectored COVID-19 vaccines indicated that only intranasal, but not intramuscular, vaccination was capable of inducing mucosal immunity against SARS-CoV-2[6,52]. Therefore, CCD is a highly effective mucosal adjuvant for protein-based vaccines, which could help build strong and comprehensive immune responses.

We previously showed that various cationic nanocarriers are potent adjuvants for a SARS-CoV-2 RBD vaccine which induced the activation of BMDCs with increased expression of maturation markers (CD40, CD86) and secretion of inflammatory cytokines[53]. Similarly, here we found that CCD/RBD-HR immunization enhanced the recruitment of CD11b$^{low}$ and CD11b$^-$ DCs in the draining lymph nodes (Fig. 4a), promoting the activation of cellular immunity[54,55]. Activation of DCs was also observed after CCD/RBD-HR treatment both in vitro and in vivo, including CD103+ DCs in lung tissues, CD11c+ DCs in lymph nodes, and BMDCs. Additionally, CCD-pre-stimulated BMDCs successfully presented antigens to T cells. All these data supported that the adjuvanticity of CCD relies on its impact on the classic APC, DCs.

The underlying mechanism of intranasal vaccines at local sites remains largely unknown. Recent studies indicated that epithelial cells may also play a pivotal role in presenting antigens under certain conditions[36,37]. As the intranasally delivered vaccines first touch numerous NECs in the nasal cavity, we hypothesized that antigens may be processed and presented by NECs. In line with our expectations, immunization with CCD/RBD-HR induced an elevation in the expression of CD40, CD80, CD86, and MHC II on NECs in vivo (Fig. 5c). NEC cell lines also showed a tendency of maturation when stimulated with CCD/RBD-HR in vitro (Fig. 5d). Similar to the results from BMDCs, NECs pre-stimulated with CCD/RBD-HR efficiently presented antigens to isolated splenic T cells (Fig. 5g), which can be abolished by a specific antibody against MHC II (Fig. 5h). In addition, as an adjuvant, CCD promoted the residue time of antigens at the mucosal surface and promoted the uptake of antigens into NECs (Fig. 5a, b). We next investigated how antigens enter NECs. Since Neu5Ac residues are ubiquitous in mammalian cells and are highly negatively charged[39–41], we speculated that the positively charged CCD may promote the interaction between antigens and Neu5Ac. Not surprisingly, pretreatment with NAs or incubation with free Neu5Ac attenuated the binding of CCD-encapsulated RBD-HR on NECs, further decreasing the presentation of antigens by NECs and reducing serum IgG responses in

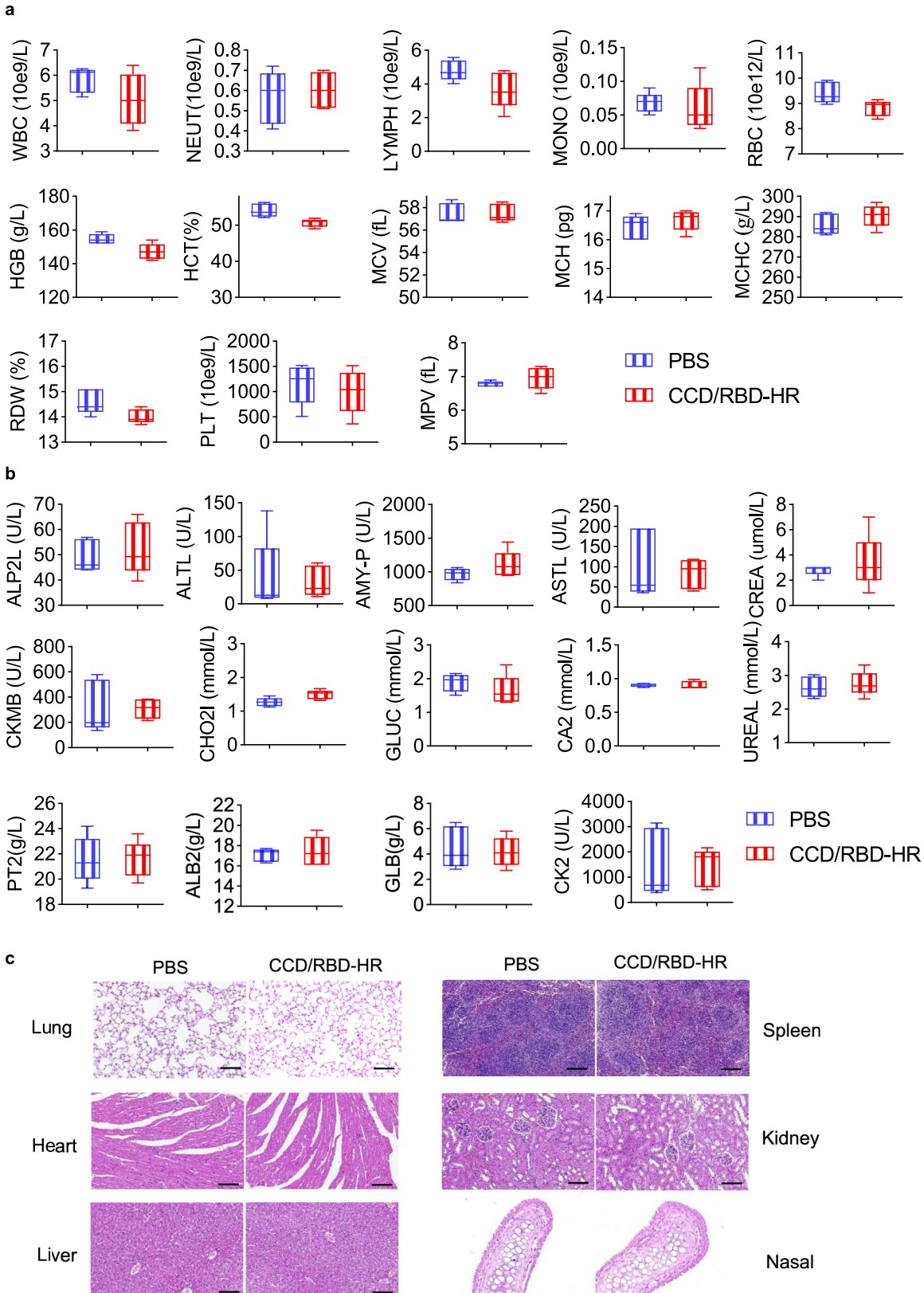

**Fig. 7 | Tolerability of CCD/RBD-HR in mice.** BALB/c mice (*n* = 5 per group) were intranasally immunized with 100 µg CCD and 10 µg RBD-HR on days 0, 14, and 28. On day 35, blood samples were collected for the assessment of complete blood count parameters (**a**) and biochemical indexes (**b**). The middle line is representative of the median, the whiskers are indicators of the value range, and the box shows the interquartile range. **c** Hematoxylin and eosin staining of the vital organs and nasal mucosa. Scale bar, 100 µm. Source data are provided as a Source Data file.

mice. Considering the huge number, NECs may play an indispensable role in the induction of immune responses for intranasal vaccines. To our knowledge, this is the first study to clarify the role of NECs as APCs for intranasal vaccines. Moreover, we determined the critical role of Neu5Ac in mediating the connections between NECs and intranasal vaccines adjuvanted by positively charged CCD for the first time.

In conclusion, we found that RBD-HR antigens adherent to CCD adjuvant can bind to NECs via Neu5Ac. This enables the residence of the intranasally delivered antigens on the nasal mucosa, providing enough time for antigen penetration in epithelial tissues. As a result, antigens can be successfully presented to T cells by local DCs and NECs, activating strong and durable immune responses against Omicron-included SARS-CoV-2 variants both locally and systemically (Fig. 6h).

## Methods

### Animals
SARS-CoV-2 viral challenge experiments were performed in the ABSL-4 facility of Kunming National High-level Biosafety Primate Research Center with approval from the Institutional Animal Care and Use Committee of the Institute of Medical Biology, Chinese Academy of Medical Sciences. All the other animal experiments in this study were performed in accordance with the guidelines approved by the Animal Care and Use Committee of Sichuan University (Chengdu, Sichuan, China). Specific pathogen-free (SPF) BALB/c and C57BL/6 mice (5–7 weeks, wild-type) were purchased from HFK bioscience company (China) and maintained at 22–23 °C, 45–55% relative humidity with a 12–12 h light-dark cycle with free food and water supplies. New Zealand rabbits were purchased from Pizhou Oriental Group. Female animals were chosen because males are aggressive and often fight and get injured, which interferes with data collection. All animals were adapted for no less than one week before experiments and were grouped randomly.

### Materials
Branched polyethylenimine (PEI 1.8 kDa) was obtained from Shanghai Aladdin Biochemical Technology Co., Ltd. N,N'-bis(2-hydroxyethyl) ethylenediamine, epichlorohydrin, and di-tert-butyl dicarbonate were purchased from Shanghai Energy Chemical Co., Ltd. Recombinant RBD protein with an Fc fragment (RBD-Fc) of SARS-CoV-2 was provided by Sino Biological. HRP-conjugated goat anti-mouse IgG, IgG1, IgG2a, IgG2b, IgG2c, IgG3, IgA antibody, and HRP-conjugated goat anti-rabbit IgG were purchased from Southern Biotech. SARS-CoV-2 pseudoviruses that highly express EGFP or luciferase were provided by Genomeditech, including the WT, P.1.617.2, B.1.1.529, BA.1, BA.2, BA.3, BA.4/5 and BA.2.12.1 pseudoviruses.

### CCD preparation
The cationic carbon dots (CCDs) consist of PEI 1.8 kDa-derived carbon dots (CD-1.8k) and a linker compound.

The preparation of CD-1.8k. Briefly, 5 g PEI 1.8 kDa was dissolved in 30 mL dried ethanol followed by the addition of 4 mL (30 wt%) $H_2O_2$ with soft stirring. The solution was transferred into a 100 mL Teflon autoclave and heated at 185 °C for 5 h. After cooling to room temperature, the mixture was filtered through a 0.45 μm filter membrane to remove insoluble solids, and the solvent was removed by reduced pressure rotary evaporation. After 24 h of vacuum drying, a thick yellow-brown oily substance was obtained and referred to as CD-1.8k.

Preparation of the CCD. The linker compound was synthesized according to our previous work[56]. Di-tert-butyl dicarbonate (40 mmol) and N,N'-Bis (2-hydroxyethyl) ethylenediamine (20 mmol) were dissolved in methanol and stirred at room temperature for 6 h. The solvent was evaporated under reduced pressure to obtain a pale yellow solid product. Then, it (6.3 mmol) was added into a mixture of epichlorohydrin (37.9 mmol), sodium hydroxide pellets (37.5 mmol), water (10 mmol), and tetrabutylammonium bromide (0.3 mmol), and stirred for 4 h at 40 °C. After that, the reaction mixture was filtered off and the solid was washed with dichloromethane. The combined organic layer was dried with anhydrous magnesium sulfate. The solvent and excess epichlorohydrins were distilled off under reduced pressure, and the residue was purified by silica gel column chromatography (v/v 4:1, PE:EA) to make the liner compound colorless oil. Then, 300 mg CD-1.8k and 150 mg linkers were dissolved in 3 mL distilled ethanol and reacted at 80 °C for 72 h under the protection of $N_2$. After solvent removal, 5 mL of trifluoroacetic acid (TFA) and 5 mL $CH_2Cl_2$ were added to the reaction mixture and stirred overnight at room temperature. The solvent was removed by vacuum evaporation, and the residue was dissolved in 3 mL of deionized water and purified by dialysis (MWCO 3500) against 0.1 N HCl solution for 2 days and deionized water for 1 day (the dialysate was changed for 12 h). Then, the products were obtained as brown oil after lyophilization, which was referred to as CCD.

### Preparation of RBD-HR
The RBD-HR antigens used in this study were prepared as previously described[5]. Briefly, the Bac-to-Bac baculovirus expression system (Invitrogen) was applied to express the trimeric RBD-HR proteins. The S-RBD sequence of the Delta variant was directly linked to the HR1 and HR2 sequences in the SARS-CoV-2 S2 subunit. The synthetic gene was amplified and incorporated into the pFastBac1 vector, which was further transformed into Escherichia coli DH10b cells for cloning. The resultant recombinant bacmids were then transfected into Sf9 insect cells (provided by our laboratory) for the production of antigens. The proteins were purified with a HisTrap Excel column (GE Healthcare) and a Superdex 200 Increase 10/300 GL column (GE Healthcare). EK protease was used to cleave the proteins during the purification step. The purity of RBD-HR was determined with SDS–PAGE and Coomassie blue staining, as well as western blotting (S antibody, Sino Biological) before use.

### Characterization of CCD and CCD/RBD-HR nanoparticles
The $^1H$-NMR spectra of PEI 1.8 kDa, CD-1.8k, linker, and CCD were measured on a Bruker AM400 NMR spectrometer with $D_2O$ as the solvent. The surface chemical stages of CCD were explored by electron spectroscopy on an AXIS Supra (Kratos Analytical, UK). The FI-IR spectra of PEI 1.8 kDa, CD-1.8k, and CCD were obtained by an IRTracer-100 (Shimadzu, Japan). CCD was mixed with RBD-HR for 30 min at 25 °C in a certain proportion with ultrapure water as the soluble substance to obtain CCD/RBD-HR nanoparticles. The suspension of the CD-1.8k, CCDs, or CCD/RBD-HR nanoparticles was dropped on a copper grid and then dried in the air, and their morphologies were observed with transmission electron microscopy (TEM, Tecnai G2 F20 S-TWIN, Thermo Fisher, USA). The zeta potentials and average hydrodynamic diameters of RBD-HR, CCD, or CCD/RBD-HR nanoparticles were measured by a Zetasizer analyzer (Malvern Instruments Ltd., UK).

### Measurement of the loading capacity of CCD/RBD-HR
The dose of RBD-HR was set as 1 μg/μL (100 μL), and CCD/RBD-HR nanoparticles with different mass ratios were obtained by incubating their mixture for 30 min at room temperature with 200 μL ultrapure water as the soluble substance. Then, the samples were placed in a 100 kDa cutoff microcentrifugal filtration tube (0.5 mL, Beyotime Biotechnology) pretreated with ultrapure water and centrifuged at 15,000×g for 60 min. The filtrate was analyzed by a BCA Protein Assay Kit (P0011, Beyotime Biotechnology) to detect the concentration of unbound proteins. The RBD-HR loading capacity was calculated using

the following equation:

$$Loading\ capacity\ (\%) = \left(1 - \frac{OD_{filtrate}}{OD_{total}}\right) \times 100\%$$

## Antigen uptake assay

6–8 weeks female BALB/c mice were sacrificed by cervical dislocation. Mouse nasal epithelial cells (NECs) were isolated from the diaphragm mucosa as described in the Supplementary information[36]. A mouse NEC cell line (CAT: BFN6021547) was purchased from BLUEFBIO. BMDCs were isolated and cultured in RPMI-1640 medium containing 10% FBS (Gibco, USA), 1% penicillin–streptomycin (Gibco, USA), 1 mM sodium pyruvate, 50 μM β-mercaptoethanol (2-ME), 20 ng/mL GM-CSF and 15 ng/mL IL-4[57]. First, we labeled the RBD-HR antigen with 488 fluorescent dyes according to the instructions of the protein labeling kit (Invitrogen). For the studies of RBD-HR uptake by NECs or BMDCs in vitro, fluorescently labeled RBD-HR (1 μg/mL) was preincubated with CCD (10 μg/mL) or PBS for 30 min at 25 °C with shaking. RBD-HR, CCDs, or CCD/RBD-HR were then coincubated with BMDCs or NECs for 30 min at 37 °C in an incubator. Cells were harvested without EDTA treatment and washed with ice-cold PBS three times. Confocal microscopy and FCM were used to determine antigen uptake.

## Coculture of BMDCs and NECs with T cells

T cells in untreated, female, 6–8 weeks BALB/c mouse spleen were obtained by a mouse CD3 T-cell enrichment kit (Stemcell). Then, T cells were labeled with fluorescence using a CFSE-cell labeling kit (Abcam) to detect T-cell proliferation. BMDCs, primary NECs (obtained from approximately 20 mice), or NEC cell lines were treated with CCD/RBD-HR (10 or 20/1 μg/mL) for 24 h at 37 °C. These treated and untreated BMDCs or NECs ($2 \times 10^5$ cells/mL) were then coincubated with CFSE-labeled T cells ($1 \times 10^6$ cells/mL) in a 37 °C incubator for 48 h. Proliferating T cells, activated T cells, and antigen-experienced T cells were then detected by FCM. In the experiment of coculturing NECs and T cells, the effect of MHC II expressed on NECs on CD4+ T cells was investigated with a neutralizing anti-mouse I-A/I-E antibody (BioLegend, 2.5 μg/mL). Rat IgG2b, κ (BioLegend) was used as an isotype control.

## Vaccinations of mice and rabbits

10 μg of purified recombinant RBD-HR protein was mixed with 100 μg of CCD and incubated at room temperature for 30 min. Then, PBS was added to a final volume of 50 μL. 6–8 week female BALB/c ($n = 5$ per group) mice were intranasally immunized with 50 μL of PBS, RBD-HR, or recombinant protein vaccine (CCD/RBD-HR) on days 0, 14, and 28. Postimmunization sera and BAL fluid were collected on days 35 and 260 after the first immunization and maintained at −80 °C until use. Female rabbits (6 months, $n = 3$ per group) were intranasally immunized with 200 μL vaccines. Rabbits in the RBD-HR group were inoculated with 20 μg RBD-HR, and those in the CCD/RBD-HR group were immunized with a complex of 20 μg RBD-HR protein formulated with 200 μg CCD adjuvants. Blood samples were collected from the marginal ear vein on day 35 after the first immunization.

## Enzyme-linked immunosorbent assay (ELISA) for antibody measurement

Antigen-specific antibodies were assayed as previously described[12]. Briefly, 96-well ELISA plates (Thermo Scientific, USA) were coated with 1 μg/mL RBD-HR antigens overnight at 4 °C, washed with PBST (PBS with 0.05% Tween-20) three times, and blocked with 0.5% BSA solution for 1 h at 37 °C. After washing plates with PBST once, serum dilutions or BAL fluid were added to the plates and incubated at 37 °C for 2 h. After being washed three times with PBST, the plates were added to 100 μL HRP-conjugated antibody (goat anti-mouse-IgG (1:10,000), IgG1

(1:10,000), IgG2a (1:10,000), IgG2c (1:10,000), IgG3 (1:10000), IgA (1:5000), and goat anti-rabbit-IgG (1:5000)) and incubated at 37 °C for 1 h. After three washes with PBST with shaking, 100 μL TMB was added to each well of the ELISA plates. After a 10 min reaction, 100 μL of stop solution was added to each well to stop the reaction. The absorbance value was read at 450 nm on a SpectraMax ABS (Molecular Devices, San Jose, CA, USA).

## ELISpot assay

Sterile 96-well ELISpot filter plates (Sigma, USA) were coated with 3 μg/mL of SARS-CoV-2 RBD protein overnight at 4 °C, washed with PBS five times, and blocked with culture medium for lymphocytes for 2 h at 37 °C. Then, lymphocytes from blood, spleen, and bone marrow were isolated with mouse lymphocyte separation medium (Dakewe Biotech Co., Ltd., China), resuspended with culture medium, and added to the plates after at least one wash. After incubation at 37 °C and 5% humidified $CO_2$ overnight, plates were washed with PBS five times and incubated with HRP-conjugated goat anti-mouse IgG (1:10000) secondary antibody (Invitrogen, USA) for 2 h at room temperature. Following additional five washes, TMB ELISpot substrate solution (Mabtech, Sweden) was added for spot formation. The reaction was terminated with rinsing water. Spots were captured and counted with IRIS FluoroSpot/ELISpot reader (Mabtech).

## Blockade of RBD binding to ACE2 receptor

In the experiment blocking the binding of RBD to ACE2, we used three RBD-Fc proteins, RBD-WT, RBD-Delta, and RBD-Omicron. The 35-day sera of each group were added to 0.3 μg/mL RBD-Fc protein in BPBS (0.1% BSA in PBS) at a 1:270 dilution and incubated for 30 min at room temperature. RBD-Fc with or without sera was subsequently added to ACE2-expressing HEK293T cells (293T/ACE2, constructed by our group[58], $5 \times 10^5$/mL) and incubated for 30 min at 4 °C. After the cells were washed with ice-cold PBS three times, PE-conjugated anti-human IgG Fc antibodies (BioLegend, USA) were added at 1:100 and stained for 30 min at 4 °C. Cells were harvested and washed three times with ice-cold PBS. The binding of RBD to ACE2 was detected with FCM.

## Live SARS-CoV-2 neutralization assay

The live SARS-CoV-2 neutralization assays were described by Yang et al.[58,59]. Briefly, Vero E6 cells (Chinese Academy of Medical Sciences and Peking Union Medical College, $5 \times 10^4$/well) were seeded in 96-well flat-bottom plates and cultured for 24 h in complete DMEM. Serum samples were inactivated by incubating at 56 °C for 30 min, and threefold serial dilutions were performed with complete DMEM. Then, 50 μL of diluted serum samples were coincubated with 50 μL of live SARS-CoV-2 virus at 100 $TCID_{50}$ (50% tissue-culture infectious dose) in a 37 °C incubator for 1 h. The serum-virus mixture was added to Vero cells and incubated for 72 h at 37 °C. A microscope was used to record the cytopathic effect (CPE), and the neutralizing titers of the immunized sera that resulted in $EC_{50}$ (50% neutralization) inhibition were calculated.

## Pseudovirus neutralization assay

In brief, equal volumes of different luciferase-expressing or EGFP-expressing pseudoviruses (WT, B1.617.2 (Delta), BA.1 (Omicron), BA.2, BA.2.12.1, BA.3, BA.4/5) were mixed with serially diluted serum solutions and incubated at 37 °C for 1 h. 293T/ACE2 cells were harvested and seeded at a density of $1.5 \times 10^5$/mL in 96-well plates. Then, a mixture of pseudovirus and serum was added to the 293 T/ACE2 cells. After the cells were cultivated in the incubator for 48 h, eGFP expression in the infected cells was detected by fluorescence microscopy and FCM. Then, the cell supernatant was removed, 50 μL PBS and 50 μL lysis reagent with luciferase substrate were added to each well, and the relative light units of the plates were read using a multimode microplate reader (PerkinElmer).

## Binding of RBD-HR protein to nasal epithelial cells

For the RBD-HR protein binding assays, CCD (30 μg/mL) was pre-incubated with Neu5Ac (Selleck; Cat: 131-48-6; 120 μg/mL) for 10 min at room temperature. His-tagged RBD-HR (10 μg/mL) was preincubated with CCD (30 μg/mL) or a mixture of CCD and Neu5Ac (CCD/Neu5Ac) for 20 min in a 37 °C incubator. Both types of NECs, including cell lines and primary NECs acquired from untreated BALB/c mice, were used in the experiment. Some NECs that had not been pretreated were incubated with the mixture (CCD/RBD-HR or CCD/RBD-HR/Neu5Ac) for 30 min at room temperature. Alternatively, some NECs were first treated with NAs (Sigma, N2876, 0.5 U/mL) overnight and then incubated with a mixture of CCD/RBD-HR. After three washes, the cells were stained with APC-conjugated anti-His antibody, and the surface-bound RBD-HR on NECs was detected by FCM.

Cell surface binding of RBD-HR could be detected by immuno-fluorescence. After the NEC cell line was incubated with the mixture (CCD/RBD or CCD/RBD-HR/Neu5Ac) or NA-treated NEC cell line was incubated with CCD/RBD-HR, all cells were fixed in 4% paraformalde-hyde and stained with human anti-S primary antibody (1:200; Sino Biological; Cat: 40150-001) and 488-anti-human secondary antibody (1:200; Proteintech; Cat: SA00003-12). Cells in the section were observed with a fluorescence microscope (Leica, Germany).

Some BALB/c mice were inoculated with CCD/RBD-HR (100/10 μg) or CCD/RBD-HR/Neu5Ac (100/10/300 μg) intranasally. Some BALB/c mice were intranasally treated with 10 μL NAs (5 U/mL) for 6 consecutive times at a one-hour interval in advance to remove the sialic acids on the NEC surface in vivo before inoculation with CCD/RBD-HR (100/10 μg) intranasally. Mouse nasal tissues acquired after 24 h were used to detect RBD-HR binding at the nasal cavity in vivo. Antibodies in the serum and BAL obtained on day 14 after immunization were determined by ELISA.

## Challenge of SARS-CoV-2 Omicron variants in BALB/c mice

Female BALB/c mice (6–8 weeks, $n = 5$ per group) were intranasally immunized with (1) PBS, (2) 100 μg CCD, or (3) 10 μg RBD-HR with 100 μg CCD in a 50 μL volume on days 0, 14, and 28 and challenged with $5 \times 10^5$ $TCID_{50}$ of SARS-CoV-2 Omicron variants intranasally. On day 4 after the challenge, the mice were euthanized, and tissues were collected. Reverse transcription-quantitative polymerase chain reaction (RT–qPCR) was used to detect viral genomic RNA (gRNA) loaded in the lung tissues, tracheas, and nasal turbinates. Specific primers and probe sets were as follows: forward, 5′-GACCCCAAAATCAGCGAAAT-3′; reverse, 5′-TCTGGTTACTGCCAGTTGAATCTG-3′; probe, 5′-FAM-ACGC CGCATTACGTTTGGTGGACCBHQ1-3′. Lung tissues of the challenged mice were collected and fixed in 10% neutral buffered formalin for at least 72 h. Then, tissues were embedded in paraffin wax and cut into 5 μm sections for standard hematoxylin and eosinite (H&E) staining.

## Cytokine and serum chemistry

Cultured supernatants of DCs and NECs were collected after the cells were stimulated for 24 h. Supernatants of T cells with DCs were collected after coculturing for 48 h at 37 °C before a 4 h treatment with brefeldin A (BFA, BioLegend). Cytokines in the supernatants, including IL-4, IFN-γ, IL-6, and others, were determined by ELISA (Invitrogen, USA) and Magnetic Luminex assay test. BALB/c mice were intranasally vaccinated with CCD/RBD-HR (100/10 μg per mouse) on days 0, 14, and 28. Blood was acquired on day 7 after the third vaccination, and blood biochemistry or blood cell counts were performed by Haiqi Biotechnology Co., Ltd. (Chengdu, China).

## Construction of fluorescently labeled RBD

Fluorescently labeled RBD was produced via biotinylation and tetra-merization as previously reported to identify RBD-binding B cells[6,60]. Briefly, streptavidin-conjugated PE (BioLegend) was added to biotinylated SARS-CoV-2 Spike RBD protein (AcroBiosystems) at a 6:1 molar ratio and incubated for 30 min at room temperature for tetramerization. The resulting RBD tetramer was used for RBD-specific B cell staining and FCM detection.

## Flow cytometry

FCM studies were carried out using a NovoCyte Flow Cytometer with NovoExpress 1.4.1 (ACEA Biosciences, Inc.).

For studies of RBD-HR uptake by NECs in vivo, 6–8 weeks, female BALB/c mice were intranasally vaccinated with 100 μg of CCD plus 10 μg of fluorescently labeled RBD-HR. After ~24 h, the nasal epithelial tissues of immunized mice were acquired and prepared into a single-cell suspension.

To detect the effect of CCD on DCs and NECs activation in vitro, BMDCs or NECs were seeded in 12-well plates ($5 \times 10^5$/well). After ~2 h, the cells were incubated with PBS, RBD-HR, or CCD/RBD-HR for 24 h in RPMI 1640 medium. Then, the cells were washed with ice-cold PBS twice and stained with antibodies.

For nasal epithelium draining cervical lymph node (CLN) cell recruitment, 6–8 weeks, female BALB/c mice were vaccinated with the vaccine (100 μg CCD/10 μg RBD-HR) via intranasal drops. 72 h after inoculation, the mice were euthanized, and CLNs were collected for detection. The lymphoid nodes were milled, treated with erythrocyte lysate for 1 min, and washed three times with ice-cold PBS to obtain a single-cell suspension. How cells are gated has been described in previous reports[32]. The different cell populations are shown in Supplementary Fig. 12a–e, including macrophages (F4/80high), monocytes (Ly6Chigh), neutrophils (Ly6Ghigh), and eosinophils (F4/80int and side scatter). Three types of DCs, including CD11blow DCs, CD11b+ DCs, and mDCs, were gated in Supplementary Fig. 12f. For RBD-specific GCB cell and Tfh cell populations, BALB/c mice were intranasally vaccinated with 100 μg CCD plus 10 μg RBD-HR on days 0, 14, and 28. Four weeks after the last immunization, CLN was obtained and detected. The gating strategy for GCB and Tfh were displayed in Supplementary Fig. 6a, b[60].

For the cell assay in BAL, vaccinated BALB/c mice were sacrificed on day 28 after the third immunization. BAL was obtained by injecting 0.5 mL ice PBS twice. Cells from BAL were harvested by centrifugation (320×g, 3 min), resuspended with RPMI 1640 medium, seeded into 12-well plates, and restimulated with peptide pools for RBD for 16 h including 4 h treatment with BFA for the analysis of T cell responses. $OVA_{257-264}$ and $OVA_{323-339}$ peptides were used as irrelevant antigen control. On day 7 after the last boost, cells in BAL were collected to detect AM responses. Data were analyzed using FlowJo software.

For the detection of CD103+ DCs and T cells in the lungs, intra-nasally vaccinated BALB/c or C57BL/6 mice (6–8 weeks, female) were sacrificed on days 7 and 28 after the third immunization, respectively. Lung tissues were first minced into extremely small pieces and digested in DMEM containing collagenase type 1 (0.5%) and collagenase type IV (0.5%) in a 37 °C incubator for one hour. The digested lung tissues were filtered with 70 mesh cell strainers, treated with red blood cell lysis buffer, and washed three times to obtain single-cell suspensions for FCM detection. Some single cells from lung tissues were restimulated with RBD peptide pools or irrelevant antigens for 16 h as mentioned before for FCM analysis.

Four weeks after the third vaccination, bone marrow was collected, lysed, washed twice, and stained with antibodies to determine RBD-specific IgG-producing plasma cells and MBCs.

## Immunofluorescence staining

For the detection of RBD-HR binding to NECs in vivo, 6–8 weeks, female BALB/c mice were intranasally immunized with CCD/RBD-HR (100 μg/10 μg per mouse) with or without pretreatment with NAs. Nasal turbinates were acquired, decalcified for 3 months, embedded in paraffin, and sectioned vertically with 3 mm thickness. Sections were fixed with 4% paraformaldehyde for 5–10 min at room temperature and blocked with 10% goat serum for half an hour. Without washing,

the sections were stained with the first antibody (SARS-CoV-2 Spike Antibody, Rabbit PAb, 1:1000, Cat: 40589-T62, Sino Biological) overnight at 4 °C and the second antibody (Goat-anti rabbit IgG, Alexa Fluor, 1:1000, Cat: A11008, Invitrogen) for 1 h at room temperature. Sections were observed with Olympus IX73 confocal microscope with CellSens standard software 2.1 (Olympus Corporation).

## Pathological evaluation of vital organs
Female BALB/c mice (6–8 weeks) were intranasally immunized on days 0, 14, and 28 and sacrificed on day 35. Vital organs (heart, liver, spleen, lung, kidney) and nosepieces were obtained and fixed in 4% buffered formalin for 72 h, embedded in paraffin, and cut into 3 μm thick slices. The slices were dewaxed in ethanol and xylene and then stained with hematoxylin-eosin (H&E). Pathology slides were digitized using Panoramic MIDI (3DHISTECH).

## Statistical analysis
All statistical analyses were performed using GraphPad software Prism 8.0. FCM data were analyzed with FlowJo V.10 and NovoExpress 1.4.1 software. $P$ values were calculated with One-way ANOVA followed by Dunnett's multiple comparisons test and Two-way ANOVA followed by Sidak's multiple comparisons tests or Tukey's multiple comparisons tests. Data are expressed as the (geometric) mean ± SEM (mean and standard error of the mean). $P$ values < 0.05 were considered significant, and ns refers to not significant.

## Reporting summary
Further information on research design is available in the Nature Portfolio Reporting Summary linked to this article.

# Data availability
All data that support the findings of this study are available with the paper and its Supplementary information. Source data are provided with this paper.

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

## Acknowledgements

This work was supported by the National Science Foundation for Excellent Young Scholars (32122052), the National Natural Science Foundation Regional Innovation and Development (No. U19A2003) and the International Cooperation Project of Sichuan Province (2021YFH0002).

## Author contributions

X.W., X.S., G.L., and S.L. conceived and supervised the research, as well as designed the experiments. H.L. prepared the vaccine, vaccinated animals, determined the titers and functions of antibodies, and analyzed the data. Xi H. prepared a CCD adjuvant and determined its interactions with RBD-HR protein. J.Y. performed the authentic SARS-CoV-2 infection in vitro and mouse challenge experiments with the Omicron variant. Z.C. formed gene cloning and expression, expressed and purified RBD-HR protein. H.H., C.H., W.R., W.H., L.C., and Xuemei H. helped with the FCM experiments. H.L. and A.A. did the rest of the experiments and wrote the manuscript. L.Y., J.L., Z.W., W.W., Y.W., and H.H. analyzed and interpreted the results, and assisted with the adjustment of directions. Y.W. and X.W. revised and edited the manuscript.

## Competing interests

The authors declare no competing interests.
