## [Peer review file · Nature Communications]

REVIEWER COMMENTS

Reviewer #1 (Remarks to the Author):

In this manuscript, Lei and colleagues describe the generation and characterization of a SARS-CoV-2 vaccine candidate comprised of carbon dot - recombinant RBD-HR proteins. The authors have recently published a report detailing the immunogenicity and protective capacity of RBD-HR against the Delta and Omicron variants of concern. Here, the novelty of the manuscript comes from the addition of cationic cross-linking carbon dots (CCD). The dots, when mixed with RBD-HR protein, spontaneously form spherical nanoparticles, which are capable of eliciting immune responses and entering cells through various mechanisms. Moreover, the CCD/RBD-HR vaccine is compatible for mucosal delivery. In general, the CCD/RBD-HR vaccine candidate, in comparison to RBD-HR alone, demonstrated an increase in all desirable traits that were evaluated. Most notably, the CCD/RBD-HR vaccine, when administered intranasally, was capable of generating immune responses that neutralized and protected mice from several variants of concern and the Omicron variant, respectively. The authors provide evidence to support the mechanism of entry and thoroughly detail the immune cell subsets that were elicited in the lymph nodes and lungs of immunized animals. Furthermore, the authors investigated the differential uptake of CCD/RBD-HR by dendritic cells and nasal epithelial cells at mucosal sites. Data from preliminary mouse toxicology studies were presented and indicated no adverse effects. Overall, the authors have provided a thorough body of work that is well presented and articulated. I have no major issues with the data or the stated conclusions. My minor comments for the authors are listed below.

- 1.) Figure 2b-j – Although perhaps a matter of stylistic preference, I would recommend making each of the symbols and legends bigger and match the color of the group in order to increase clarity for the reader.
- 2.) Figure 2 - The authors should provide and define limits of detection for many of the panels.
- 3.) Figure 2b-j – The authors should perform statistical analyses between the groups and indicated if they are statistically significant.
- 4.) Can the authors provide (perhaps in the supplemental data) the raw cell numbers for Figure 2k-p.
- 5.) Figure 2o – the y-axis label “C”D8 is cutoff.

6.) Figure 5a – the images are quite dim, making interpretation of the data difficult.

7.) The authors need to clarify whether the NEC lines are primary cells or transformed. In either case, additional details should be provided in the methods and/or figure legends to clarify the source and methods of generation of the cells for the reader.

8.) Have authors considered dosing down the 100ug CCD/ 10ug RBD-HR dose given to mice? This data would have important implications on potentially efficacious doses for humans and potential scalability of the vaccine.

Reviewer #2 (Remarks to the Author):

This paper is exciting and presents research that appears to be quite promising with regard to an intranasal strategy to elicit immunity to respiratory pathogens and engage antigen presenting cells. The descriptions of the physical parameters are excellent. Also, the evidence for protective and long-lived antibody responses is strong and compelling as is the evidence for sterilizing immunity via antibodies after vaccination.

However, the cellular assays are problematic at a number of levels, described below.

1. Very little evidence of antigen specificity in the cellular assays is provided by the authors. Even the one *in vitro* culture for T cells and cytokine is very atypical and involves a 3 day culture, no irrelevant antigen controls and no measurement of frequency of antigen reactive cells directly *ex vivo*, which is standard. Direct antigen specific stimulation and quantification, with control antigens is essential for any assessment or conclusions about recruitment of T cells.

2. The same concerns regarding antigen specificity persists with the figures that show Tfh, GC cells in the CLN, or T cells in the BAL, or B and T cells in the spleen do not distinguish recruitment of antigen specific cells or influx of irrelevant cells because of generally elicitation of inflammatory mediators. There are reagents available to detect antigen specific T cells and B cells and these should be implemented.

3. The time between the final boost of 3 vaccinations is very short, and so no comment or conclusions about “memory or “tissue resident memory” can be made. See line 92.

4. CD44 is not a marker of memory cells-it simply indicates "antigen-experience" and is found to be upregulated soon after initial priming and persists. It distinguishes naïve from antigen experienced cells.

5. In general, most assays involve measurements done very soon after a 3rd vaccination. 3 vaccinations are a lot and any clinically valuable vaccine needs to elicit cells that persist for at least 30 days. Also, many irrelevant cells in the host can be recruited transiently in the respiratory tract and these limited time spans do not allow distinction between antigen dependent and independent recruitment of cells.

We thank the reviewers for their generous comments on the manuscript. To address their concerns, we edited the manuscript and wrote a point-to-point response as follows. Please note that we have also **yellow-highlighted** the amendments in the marked-up version of the revised manuscript to help the reviews find and understand our changes.

Responses to the reviewer's comments:

Reviewer #1:

1. **Response to comment:** *“Figure 2b-j – Although perhaps a matter of stylistic preference, I would recommend making each of the symbols and legends bigger and match the color of the group in order to increase clarity for the reader.”*

Response: Thank you for your suggestion. We have enlarged the symbols and legends and matched the color of different groups in Fig. 2b-j.

2. **Response to comment:** *“Figure 2 - The authors should provide and define limits of detection for many of the panels.”*

Response: Thank you for your suggestion. We have added the limits of the detection as dotted lines in Fig. 2b-f, Fig. 2i, j, and Supplementary Fig. 2.

3. **Response to comment:** *“Figure 2b-j – The authors should perform statistical analyses between the groups and indicated if they are statistically significant.”*

Response: Thank you for pointing out this issue. We have provided the exact p values in Fig. 2b-j.

4. **Response to comment:** *“Can the authors provide (perhaps in the supplemental data) the raw cell numbers for Figure 2k-p.”*

Response: We sincerely thank the reviewer for this suggestion. In the revised manuscript, we have added some experiments and changed the original Figure 2k-p with new data in Figure 2o-s. We have provided the raw cell numbers for

Figure 2o-s. in Supplementary Figure 4d, e and Supplementary Fig. 5b, c in the latest version.

5. Response to comment: *“Figure 2o – the y-axis label “C”D8 is cutoff.”*

Response: Thank you for pointing out this mistake. We have modified some experiments and the original Fig, 2o was replaced with Fig. 2q in the revised manuscript, where we have corrected the y-axis label.

6. Response to comment: *“Figure 5a – the images are quite dim, making interpretation of the data difficult.”*

Response: Thank you for your suggestion. We have modified Figure 5a to make it look brighter.

7. Response to comment: *“The authors need to clarify whether the NEC lines are primary cells or transformed. In either case, additional details should be provided in the methods and/or figure legends to clarify the source and methods of generation of the cells for the reader”*

Response: Thank you for your suggestion. We have added the description about the sources of different NECs as follows:

“NEC” was changed into “primary NECs”: line 276 and 280;

“NEC” was changed into “isolated primary NECs”: line 292;

“NEC” was changed into “NEC cell lines”: line 392;

“NECs (obtained from approximately 20 mice)” was changed into “primary NECs (obtained from approximately 20 mice) or NEC cell lines”: line 491-492

We also added the sources of the NEC cell line in the Methods as “A mouse NEC cell line was purchased from BLUEFBIO” (Line 479-480). The methods of isolating primary NEC were provided in the *“Isolation and culture of NECs”*

section in Methods of Supplementary Material (Line 27-34).

- 8. Response to comment:** “Have authors considered dosing down the 100ug CCD/10ug RBD-HR dose given to mice? This data would have important implications on potentially efficacious doses for humans and potential scalability of the vaccine.”

Response: Thanks a lot for this advice. In fact, we have investigated the efficacy of our vaccine with different doses (CCD/RBD-HR: 25µg/2.5µg, 50µg/5µg, 100µg/10µg). As shown in the following figure, the IgG antibody titers increase with the increase of vaccine dose in mice on day 14. Given the safety concerns, we selected 100µg/10µg as the optimal dose in the following experiments.

Fig. The IgG titers induced by different doses of CCD/RBD-HR on day 14 in mice.

Reviewer #2:

- 1. Response to comment:** “Very little evidence of antigen specificity in the cellular assays is provided by the authors. Even the one in vitro culture for T cells and cytokine is very atypical and involves a 3 day culture, no irrelevant antigen controls and no measurement of frequency of antigen reactive cells directly ex vivo, which is standard. Direct antigen specific stimulation and quantification,

with control antigens is essential for any assessment or conclusions about recruitment of T cells.”

Response: We thank the reviewer for the suggestion and have done so in our revised manuscript. For one hand, lymphocytes from lung, spleen, and BAL were stimulated with peptide pools for RBD for 12 h before 4 h treatment with brefeldin A. For another, an irrelevant antigen control, namely OVA₂₅₇₋₂₆₄ and OVA₃₂₃₋₃₃₉ peptides for the model antigen OVA, was used to stimulate T cells at the same time. Finally, the frequencies of activated (CD69), antigen-experienced (CD44), and cytokine (IFN- γ and IL-4) secreting cells were assessed with flow cytometry. The results were displayed in Fig. 2q-s, Fig. 3b, c, and Fig. 3g. We have added the description about the results as well. We presented this part of data as follows:

Line 164-174: Next, splenic CD3⁺ T cells from the immunized mice were isolated and stimulated *ex vivo* with overlapping peptide pools for RBD. Peptides for the model antigen OVA (OVA₂₅₇₋₂₆₄ and OVA₃₂₃₋₃₃₉) were used as irrelevant antigen control. After restimulation with RBD peptides rather than OVA peptides, antigen-experienced (CD44⁺) and activated (CD69⁺) CD4 T⁺ and CD8 T⁺ cells in the spleen were significantly elevated in mice vaccinated with CCD/RBD-HR (Fig. 2q, r, Supplementary Fig. 5a, b). The production of interferon- γ (IFN- γ) and interleukin-4 (IL-4) was determined by intracellular cytokine staining. We observed a higher proportion of splenic CD4 T⁺ and/or CD8 T⁺ cells expressing IL-4 and IFN- γ in mice inoculated with CCD/RBD-HR compared with pure RBD-HR immunization (Fig. 2s, Supplementary Fig. 5c), demonstrating the activation of antigen-specific T-cell immunity systemically.

Line 182-189: Then we also tested the antigen-specific T cell response of the lung after immunization. Lung tissues were harvested four weeks post-boosting and T cells were evaluated with FCM. Restimulation with a pool of RBD peptides induced a remarkable elevation in the frequency of IFN- γ -generating CD8⁺ and CD4 T⁺ cells in the lungs of mice receiving CCD/RBD-HR vaccine (Fig. 3b, c, Supplementary Fig. 7). In contrast, irrelevant antigen restimulation

showed minimal effects on IFN- γ producing, indicating the induction of antigen-specific T cell responses.

Line 218-221: We also determined antigen-specific T cells in mononuclear cells of the BAL with FCM for intracellular IFN- γ and TNF- α expression upon *ex vivo* restimulation with peptide pools for RBD or OVA peptides control. As expected, CD8⁺ T cells from CCD/RBD-HR-immunized mice responded to peptide pools for RBD in four weeks after the last boosting (Fig. 3g).

The details of the methods have been added in the Methods in the revised manuscript (Line 649-654 and 661-663).

2. **Response to comment:** *“The same concerns regarding antigen specificity persists with the figures that show Tfh, GC cells in the CLN, or T cells in the BAL, or B and T cells in the spleen do not distinguish recruitment of antigen specific cells or influx of irrelevant cells because of generally elicitation of inflammatory mediators. There are reagents available to detect antigen specific T cells and B cells and these should be implemented”*

Response: Thank you for your suggestion. In the revised version of our manuscript, the specificity of T cell responses in the BAL and spleen has been proved as mentioned in response to the 1st comments. We have made two major modifications of our experiments to determine antigen-specific B cells responses.

First, ELISpot analysis with RBD proteins was used to evaluate the induction of RBD-specific IgG antibody-secreting cells in blood, spleen, and bone marrow upon vaccination as previously reported¹. The results were displayed in Fig. 2k-m. The results have been presented as “ELISpot analysis with RBD protein indicated that CCD/RBD-HR immunization induced high levels of RBD-specific IgG antibody-secreting cells (ASCs) in blood, bone marrow and spleen, whilst the control RBD-HR did not (Fig. 2k-m) (Line 154-156).” The protocol of ELISpot assay was added in the Methods (526-536).

Second, A fluorescently labelled RBD protein was produced to gate out RBD-binding B cells with flow cytometry^{2,3} (Supplementary Fig. 4 and Supplementary Fig. 6b). We used this labelled protein to detect RBD-binding GCB in the CLN (Fig. 3a), IgG-producing CD138⁺ plasma cells (Fig. 2o), and memory B cells (Fig. 2p). The results were interpreted as “A fluorescently labelled RBD protein was produced to gate out RBD-binding B cells (Supplementary Fig. 4a). Similarly, we found that CCD/RBD-HR immunization resulted in an elevation in the frequency of RBD-specific IgG-producing CD138⁺ plasma cells (PCs) in the circulation and bone marrow with FCM (Fig. 2o, Supplementary Fig. 4b-d), further supporting the activation of strong B-cell responses. Memory B cells (MBCs, CD38⁺CD80⁺) can enhance rapid recall responses upon secondary exposure⁴. As a mucosal adjuvant, CCD substantially promoted the expansion of RBD-specific MBCs in the blood, bone marrow, and spleen (Fig. 2p, Supplementary Fig. 4e), accounting for the sustainability of humoral immunity. (Line 156-164)” and “.....antigen-specific germinal center B (GCB, GL7⁺CD95⁺RBD⁺) cells in the CLN was observed in the CCD/RBD-HR group compared with immunization with RBD-HR alone (Fig. 3a, Supplementary Fig. 6a, b).....(Line 180-182)”. The construction of the labelled RBD protein was detailed in the Methods (Line 620-625).

However, the detection of antigen-specific Tfh requires the generation of antigen-specific MHC II multimers. According to Proimmune (the most prestigious company for multimer design and production) and previous reports^{5,6}, there is only one multimer available for the detection of CD4⁺ T cell responses in COVID-19 currently, which is designed for human samples (peptide sequence: QALNTLVKQLSSNFGAI), making it impossible for testing Tfh cells in mice. Therefore, we sincerely apologize for our inability to provide the results in this part in such a short time. Nevertheless, since our added results have comprehensively proved the generation antigen-specific T and B cell responses in multiple organs and tissues, we believe the enhanced Tfh responses upon

immunization is antigen-specific rather than antigen-independent. And we will keep investigating this area in our future work.

3. **Response to comment:** *“The time between the final boost of 3 vaccinations is very short, and so no comment or conclusions about “memory or “tissue resident memory” can be made. See line 92.”*

Response: Thank you for this precious idea. We extended the detection of cellular immunity to 4 weeks after the third immunization (Fig. 2a). The details will be discussed in the response to comment 5.

4. **Response to comment:** *“CD44 is not a marker of memory cells-it simply indicates “antigen-experience” and is found to be upregulated soon after initial priming and persists. It distinguishes naïve from antigen experienced cells.”*

Response: Thank you for your suggestion. We have corrected “memory cells” into “antigen-experienced cells” on lines 167, 199, 201, 215-216, 281, 312-313, 494-495, 888, 907, 913.

5. **Response to comment:** *“In general, most assays involve measurements done very soon after a 3rd vaccination. 3 vaccinations are a lot and any clinically valuable vaccine needs to elicit cells that persist for at least 30 days. Also, many irrelevant cells in the host can be recruited transiently in the respiratory tract and these limited time spans do not allow distinction between antigen dependent and independent recruitment of cells.”*

Response: Thanks again for your suggestion. In order to test the persistence of cellular immunity in the lung, BAL, blood, bone marrow, CLN, and spleen, mice were sacrificed four weeks after the last immunization. We found that the tendency was the same as before. Since the data from day 56 are more meaningful than that from day 35, we have replaced/added the data about cellular immunity with the latest ones (Fig. 2k-s, Fig. 3a-d, Fig. 3f-g, Supplementary Fig. 4-7, Supplementary Fig. 9, 10). We also changed the description about the

detection time point on line 152, 178, 185, 200, 212, 221, 359, 646, 664, 883, and 904.

To discriminate antigen dependent and independent recruitment of cells, many experiments have been modified to determine antigen-specific cellular response as mentioned in the responses to Comment 1 and 2.

1. Hassan, A.O. et al. A Single-Dose Intranasal ChAd Vaccine Protects Upper and Lower Respiratory Tracts against SARS-CoV-2. *Cell* **183**, 169-184. e113 (2020).
2. Afkhami, S. et al. Respiratory mucosal delivery of next-generation COVID-19 vaccine provides robust protection against both ancestral and variant strains of SARS-CoV-2. *Cell* **185**, 896-915. e819 (2022).
3. Alameh, M.G. et al. Lipid nanoparticles enhance the efficacy of mRNA and protein subunit vaccines by inducing robust T follicular helper cell and humoral responses. *Immunity* **54**, 2877-2892. e2877 (2021).
4. Sallusto, F., Lanzavecchia, A., Araki, K. & Ahmed, R. From vaccines to memory and back. *Immunity* **33**, 451-463 (2010).
5. Peng, Y. et al. Broad and strong memory CD4(+) and CD8(+) T cells induced by SARS-CoV-2 in UK convalescent individuals following COVID-19. *Nat Immunol* **21**, 1336-1345 (2020).
6. Jung, S. et al. The generation of stem cell-like memory cells early after BNT162b2 vaccination is associated with durability of memory CD8(+) T cell responses. *Cell Rep* **40**, 111138 (2022).

REVIEWERS' COMMENTS

Reviewer #1 (Remarks to the Author):

Thank you to the authors for their responses. My concerns have been adequately addressed.

Reviewer #2 (Remarks to the Author):

This is a much improved paper that illustrates the promise of a novel cationic crosslinked carbon dot intranasal vaccine to initiate protective immunity in mice. Comprehensive analyses on B and T cell responses were performed that now includes antigen specificity addressed.

The issues of concern raised in my initial review were satisfied by the additional data provided in the revised manuscript.